# Congenital disorder of glycosylation caused by starting site-specific variant in syntaxin-5

Peter T. A. Linders [1], Eveline C. F. Gerretsen [1], Angel Ashikov [2,3], Mari-Anne Vals[4,5], Rinse de Boer[6], Natalia H. Revelo[1], Richard Arts[1], Melissa Baerenfaenger[2], Fokje Zijlstra[3], Karin Huijben[3], Kimiyo Raymond[7], Kai Muru[5,8], Olga Fjodorova[8], Sander Pajusalu[5,8], Katrin Õunap[5,8], Martin ter Beest[1], Dirk Lefeber[2,3 ✉] & Geert van den Bogaart [1,9 ✉]

The SNARE (soluble N-ethylmaleimide-sensitive factor attachment protein receptor) protein syntaxin-5 (Stx5) is essential for Golgi transport. In humans, the *STX5* mRNA encodes two protein isoforms, Stx5 Long (Stx5L) from the first starting methionine and Stx5 Short (Stx5S) from an alternative starting methionine at position 55. In this study, we identify a human disorder caused by a single missense substitution in the second starting methionine (p.M55V), resulting in complete loss of the short isoform. Patients suffer from an early fatal multisystem disease, including severe liver disease, skeletal abnormalities and abnormal glycosylation. Primary human dermal fibroblasts isolated from these patients show defective glycosylation, altered Golgi morphology as measured by electron microscopy, mislocalization of glycosyltransferases, and compromised ER-Golgi trafficking. Measurements of cognate binding SNAREs, based on biotin-synchronizable forms of Stx5 (the RUSH system) and Förster resonance energy transfer (FRET), revealed that the short isoform of Stx5 is essential for intra-Golgi transport. Alternative starting codons of Stx5 are thus linked to human disease, demonstrating that the site of translation initiation is an important new layer of regulating protein trafficking.

[1] Department of Tumor Immunology, Radboud Institute for Molecular Life Sciences, Radboud University Medical Center, 6525 GA Nijmegen, The Netherlands. [2] Department of Neurology, Donders Institute for Brain, Cognition and Behavior, Radboud University Medical Center, 6525 GA Nijmegen, The Netherlands. [3] Translational Metabolic Laboratory, Department of Laboratory Medicine, Radboud University Medical Center, 6525 GA Nijmegen, The Netherlands. [4] Children's Clinic, Tartu University Hospital, Tartu, Estonia. [5] Department of Clinical Genetics, Institute of Clinical Medicine, University of Tartu, Tartu, Estonia. [6] Molecular Cell Biology, Groningen Biomolecular Sciences and Biotechnology Institute, University of Groningen, Groningen, The Netherlands. [7] Department of Laboratory Medicine and Pathology, Mayo College of Medicine, Rochester, MN, USA. [8] Department of Clinical Genetics, United Laboratories, Tartu University Hospital, Tartu, Estonia. [9] Department of Molecular Immunology, Groningen Biomolecular Sciences and Biotechnology Institute, University of Groningen, 9747 AG Groningen, The Netherlands. ✉email: dirk.lefeber@radboudumc.nl; g.van.den.bogaart@rug.nl

In eukaryotes, proteins destined for the secretory pathway are synthesized at the endoplasmic reticulum (ER) and then transported to the Golgi apparatus, where they are sorted for their ultimate destinations at the *trans*-Golgi network (TGN). Central to this process is intracellular membrane fusion, which is mediated by members of the SNARE (soluble *N*-ethylmaleimide-sensitive factor attachment protein receptor) protein family. Cognate SNARE proteins that are present in both the carrier vesicle (v-) and target (t-) membranes, called v- and t-SNAREs, respectively, engage and form a tight alpha-helical coiled-coil bundle that overcomes the energy barrier of membrane fusion. Membrane fusion requires a single R-SNARE, characterized by an arginine residue located central in the SNARE bundle, and three Q-SNAREs, with glutamine residues instead. Generally in mammalian cells, the R-SNAREs act as v-SNAREs and the Q-SNAREs together form the t-SNARE complex on the target membrane[1]. In contrast, the Qc-SNAREs Bet1 and Bet1L (GS15) function as v-SNAREs at the ER/Golgi interface[2–5], while, for anterograde ER-to-Golgi trafficking, the recipient t-SNARE complex is formed by the Qa-SNARE syntaxin-5 (Stx5)[6–9], together with the Qb-SNAREs GosR1 (also known as GS27 or membrin) or GosR2 (GS28), and R-SNAREs Ykt6 or Sec22b (Ers24)[10–12]. This different allocation of the Qc-SNAREs Bet1 and Bet1L as v-SNARE instead of t-SNARE possibly prevents the formation of non-functional SNARE complexes during ER-to-Golgi transit. In addition, Stx5 functions in retrograde intra-Golgi transport by forming a recipient t-SNARE complex with GosR1, and forms a complex with Ykt6[5,13] for retrograde trafficking from endosomes to the TGN[14,15], making it a unique SNARE protein involved in both anterograde and retrograde Golgi transport.

*STX5* is highly conserved and is an essential gene in animals and fungi[16,17]. In animals, Stx5 exists as a long and a short isoform translated from the same mRNA: 39.6 kDa-sized Stx5 Long (Stx5L) and 34.1 kDa-sized Stx5 Short (Stx5S)[13,18]. This is in contrast to lower organisms, such as *Saccharomyces cerevisiae*, which only express a single isoform of Stx5 (Sed5p). Although Sed5p was originally believed to resemble mammalian Stx5S[13], it is now clear that it likely more resembles Stx5L, since an N-terminal COPI-binding tribasic motif has been identified in Sed5p[19]. The emergence of a second Stx5 isoform can be traced back to the pacific purple sea urchin, *Strongylocentrotus purpuratus*, and is also present in the model organism *Danio rerio*, but not in *Drosophila melanogaster* nor *Caenorhabditis elegans*. Compared to Stx5S, Stx5L contains 54 extra N-terminal residues bearing an Arginine–Lysine–Arginine (RKR) ER-retrieval motif, and as a result, Stx5L locates more to the ER, whereas Stx5S locates more to the Golgi network[13,18,20–22]. The evolutionary necessity of the two Stx5 isoforms remains unclear but it has been suggested that Stx5L is important to maintain ER structure by binding microtubules, possibly via CLIMP-63[21,23]. In addition, immunoprecipitations showed that GosR1 and Bet1L preferentially interact with Stx5S over Stx5L[24,25], suggesting that Stx5S might act in more fusogenic complexes later at the ER–Golgi interface, whereas Stx5L might be more involved in earlier fusion steps.

In the present study, we identified a genetic variant in the second translation codon methionine-55, fully abrogating the production of Stx5S and providing an opportunity to study the physiological relevance of the existence of two isoforms in humans. Patients homozygous for this mutation have a severe clinical phenotype associated with infantile mortality and defective protein glycosylation. We demonstrate that although Stx5L can largely compensate for the lack of Stx5S, the loss of Stx5S leads to defects in intra-Golgi trafficking with altered morphology of the ER and Golgi apparatus and mislocalization of glycosyltransferases, which results in pronounced defects in glycosylation.

Moreover, by synchronizing the intracellular trafficking of Stx5 isoforms and Förster resonance energy transfer microscopy (FRET) coupled to fluorescence lifetime imaging microscopy (FLIM), we reveal differential SNARE interactions for either isoform and identify Stx5S as the dominant Qa-SNARE for intra-Golgi transport. A mutation in an alternate starting site of ribosomal translation of Stx5 is thus linked to human disease. This finding reveals that protein function can be regulated at the level of translation initiation and has profound effects on intracellular membrane trafficking and Golgi function.

## Results

**Clinical data**. The family history (Supplementary Fig. 1) revealed multiple deceased individuals (IV:3, IV:9, IV:10) shortly after birth, spontaneous abortions (IV:5, IV:6, IV:7), and elective abortions in the 20th–21st week of pregnancy due to abnormal fetal ultrasound (US) (IV:4, IV:8). Fetal US of individuals IV:8, IV:9, and IV:10 showed shortening of the long bones with suspicion of chondrodysplasia. Patients IV:9 and IV:10 showed highly dysmorphic facial features (high forehead, frontal bossing, prominent glabella, short and upturned nose, long philtrum, micrognathia, and dysplastic ears), skeletal dysplasia (short extremities and narrow thorax), profound hypotonia, hepatomegaly, and many abnormal laboratory parameters including elevated cholesterol (Supplementary Table 1). After birth, the main clinical problem for both patients IV:9 and IV:10 was progressive liver failure with cholestasis and hyperinsulinemic hypoglycemia (Supplementary Table 1). Liver failure was the main cause of death at the age of 28 days and 8 months, in patients IV:9 and IV:10, respectively. Autopsy of fetus IV:8 revealed bilateral hydronephrosis and sacral lordosis. Autopsy of patient IV:9 showed hepatomegaly with stage 3–4 liver fibrosis, agenesis of left kidney, hyperemia of internal organs, ventricular septal defect, and suggestive pathohistological features of chondrodysplasia. Autopsy of patient IV:10 showed biliary cirrhosis and nodular regenerative hyperplasia, pancreatic hypertrophia/hyperplasia, and narrow thorax with normal lung development.

**Abnormal protein glycosylation suggests a defect in Golgi trafficking**. Known genetic causes for skeletal dysplasias were excluded (IV:8), no submicroscopic chromosomal abnormalities were found, while most metabolic investigations were normal except for the congenital disorders of glycosylation (CDG) (IV:9 and IV:10). CDG screening revealed a strong hyposialylation of protein *N*-glycosylation and mucin-type *O*-glycosylation, as analyzed by isofocusing of, respectively, plasma transferrin (Fig. 1a and Supplementary Table 2) and apolipoprotein CIII (ApoCIII-IEF, Fig. 1b and Supplementary Table 2). ApoCIII-IEF showed a strong increase of non-sialylated apoCIII (ApoCIII-0) band intensities compared to the intensities of the fully glycosylated form, even stronger than those observed for genetic defects in the conserved oligomeric Golgi (COG) complex, a known group of disorders with disturbed Golgi homeostasis and abnormal glycosylation[26].

To gain more insight into the abnormal *N*-glycan structures, mass spectrometry was performed of intact transferrin (Fig. 1c, d and Supplementary Fig. 2) and of total plasma protein-derived *N*-glycans (Fig. 1c, e and Supplementary Fig. 3). Analysis of intact transferrin of individuals IV:9 and IV:10 revealed multiple abnormal glycan structures, divided into two categories: high-mannose structures and truncated glycans. Compared to the peak height of completely glycosylated transferrin, a dominant accumulation was found of high-mannose glycans (Supplementary Table 3, Man5, mass/peak number) suggesting a problem with MGAT1, the enzyme that adds the next *N*-acetylglucosamine during *N*-glycosylation. Furthermore, a

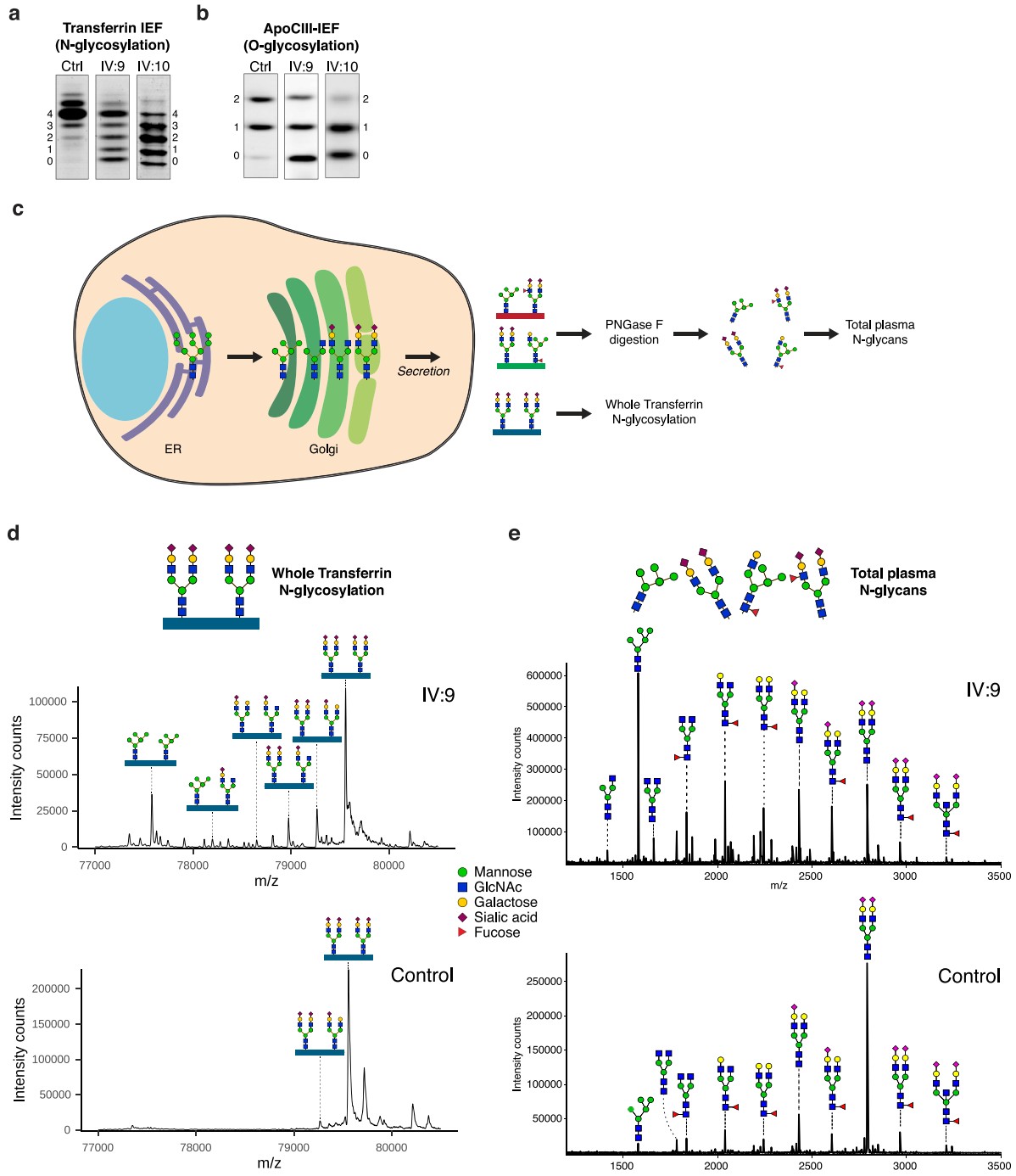

**Fig. 1 A novel, lethal, genetic variant suggests a defect in protein glycosylation related to Golgi trafficking. a** Glycosylation screening by isoelectric focusing (IEF) of serum transferrin. The accompanying numbers represent the total number of sialic acids in the different proteoforms. Both patients show a reduction in the number of sialic acids. Quantification of bands is shown in Supplementary Table 2. **b** Glycosylation screening by IEF of serum apolipoprotein C3 (ApoCIII). ApoCIII has one mucin-type O-linked glycan with one or two sialic acids in controls. Both patients show a reduction in the number of sialic acids. Quantification of bands is shown in Supplementary Table 2. **c** Schematic overview of N-glycosylation intermediates in the Golgi. For mass spectrometry analysis of glycan structures, glycosylated transferrin was enriched from all secreted glycoproteins in human serum and subjected to intact protein mass spectrometry. In parallel, a different serum sample was treated with PNGase F to cleave and analyze N-glycans from all plasma proteins. **d** Nanochip-C8 QTOF mass spectra of enriched intact serum transferrin of Stx5M55V patient IV:9 (top spectrum) and healthy control (lower spectrum). Key transferrin glycoforms are shown, indicating a strong increase of high-mannose glycans and glycans lacking sialic acid and galactose. Annotation of all peaks of patients IV:9 and IV:10 is shown in Supplementary Table 3. **e** MALDI-TOF mass spectra of total plasma N-glycans of Stx5M55V patient IV:9 (top spectrum) and healthy control (lower spectrum). Structural analysis shows a strong increase of high-mannose glycans and glycans lacking sialic acid and galactose. Annotation of all peaks of patients IV:9 and IV:10 is shown in Supplementary Table 4.

series of transferrin isoforms was observed with reduced incorporation of galactose and sialic acid residues. Similar glycosylation patterns were observed in COG5-CDG[27–30]. Analysis of *N*-glycans released from total plasma proteins recapitulated the two categories of abnormal glycans with the accumulation of high-mannose glycans, as well as reduced incorporation of galactose and sialic acid residues (Supplementary Table 4). Together, these data indicate that the activities of multiple glycosyltransferases in the Golgi apparatus are affected, covering both *N*- and *O*-glycosylation, thereby suggesting a general disturbance in Golgi trafficking.

**Molecular investigations result in the identification of variants in STX5.** Chromosomal microarray analysis (CMA) using HumanCytoSNP-12 microarrays revealed multiple long contiguous stretches of homozygosity (LCSH, >5 Mb) distributed across the entire genome, with several regions of homozygosity on chromosome 11 in all three affected sibs (IV:8, IV:9, and IV:10, Supplementary Table 5 and Supplementary Data 1). Exome sequencing was performed in proband IV:9 to find the genetic variant that could be associated with the disease. Only two homozygous rare protein-altering variants without homozygous individuals in the gnomAD v3 database were identified in shared homozygous stretches on chromosome 11. First, a missense variant in the *VPS37C* gene was discovered (NM_017966.4:c.760 G > T p.(Gly254Cys) rs201088253). However, as this variant reaches an allele frequency of 0.9% in Estonia, it is unlikely to cause a rare genetic disorder. The second variant was identified in the *STX5* gene (NM_003164.4:c.163 A > G p.(Met55Val), Fig. 2a). This is a missense mutation affecting the alternative starting codon for the production of the short Stx5 isoform. The variant is absent from the gnomAD v3 database, and was thus classified as a potentially disease-causing variant. The variant was confirmed by Sanger sequencing as homozygous in all the affected individuals (IV:8, IV:9, and IV:10) and as heterozygous in their mother (III:2). Paternal DNA was not available for testing.

To confirm the effect of the genetic variant on both Stx5 proteoforms, immunoblotting was performed in primary dermal fibroblasts of patients IV:9 and IV:10. While Stx5L was present, a total absence of Stx5S was found in both patient fibroblasts (Fig. 2b, c). We next tested the expression of known interaction partners of Stx5. The levels of Qc-SNARE Bet1L, which forms a complex with Stx5 upon retrograde intra-Golgi trafficking[5,13–15], were also reduced. In contrast, the expression of Qc-SNARE Bet1, which forms a complex with Stx5 upon anterograde ER–Golgi trafficking[10,12,13,31], was not reduced (Fig. 2b, c). Likewise, the expression of Qb-SNAREs GosR1 and GosR2, which can complex with Stx5 for anterograde ER–Golgi trafficking and retrograde intra-Golgi trafficking[5,9,10,12,13,24], were unaltered in patient dermal fibroblast lysates. We hypothesized that a compensatory mechanism might exist by upregulating the expression of the *trans*-Golgi Qa-SNARE Stx16[32], usually involved in endosome-to-TGN trafficking[33], but we did not detect a change of Stx16 expression in patient fibroblast lysates (Fig. 2b, c). As a first step to confirm that fibroblasts offer a useful model to recapitulate the cell biological abnormalities due to loss of the Stx5S isoform, we studied glycosylation by fluorescently labeled lectins.

**Glycosylation defects in Stx5M55V patient fibroblasts.** Patient fibroblasts could be cultured normally and we did not observe apparent differences in growth rate or viability between healthy control and patient-derived cell lines. Cell-surface staining with the lectin SNA-I from *Sambucus nigra*, which binds terminal sialic acid in an α−2,6 linkage of fully formed *N*-glycan moieties and, to a lesser extent, sialic acid in an α−2,3 linkage, showed that glycosylation was also impaired at the cellular level in patient fibroblasts. Compared to fibroblasts of healthy donors, we

observed a more than two-fold reduced SNA-I-labeling intensity in Stx5M55V patient fibroblasts (Fig. 2d, e). Moreover, most signal came from punctuated structures in the Stx5M55V patient fibroblasts, instead of the more uniform cell membrane labeling observed in the healthy donor fibroblasts. To confirm this glycosylation defect, we performed cell-surface staining with the lectin PNA (Peanut agglutinin) from *Arachis hypogaea*, which binds terminal galactose residues present on mucin *O*-glycan moieties of incompletely glycosylated proteins. Opposite to our findings with SNA-I, we observed an increased labeling intensity in Stx5M55V patient fibroblasts relative to healthy control by about six-fold (Fig. 2f, g). These findings show that patient-derived fibroblasts, which express Stx5L but lack Stx5S, have a glycosylation defect.

Next, we evaluated the expression levels of Stx5S and Stx5L in lysates of peripheral blood mononuclear cells (PBMCs) obtained from four different healthy donors (Supplementary Fig. 4). While total Stx5 expression levels varied strongly (more than two-fold), the ratio of Stx5S and Stx5L was approximately equimolar for all individuals, demonstrating that the expression ratio of the two isoforms is similar in PBMCs of different healthy subjects.

To determine the role of Stx5S in Golgi transport, we generated a Stx5L-lacking fibroblast cell line from one of the prior-used control lines, using CRISPR/Cas9 (Fib Stx5ΔL, Supplementary Fig. 5a). We then performed the same cell-surface staining as described above. While we observed a similar decrease in SNA-I-labeling intensity in Stx5ΔL fibroblasts as in Stx5M55V patient fibroblasts, the staining pattern was more similar to healthy control and showed a uniform plasma membrane-localized SNA-I labeling (Supplementary Fig. 5b, c). In addition, opposite to Stx5M55V fibroblasts, the low PNA-labeling intensity was reduced further in Fib Stx5ΔL (Supplementary Fig. 5d, e). To investigate this difference further, we generated two clonal HeLa cell lines lacking Stx5L using the same method (Stx5ΔL: B1A7 and C1F4, Supplementary Fig. 5f), and we observed a decrease in the already low SNA-I-labeling intensity as measured by FACS compared to the parental HeLa cells, while PNA labeling resulted in opposite changes between the two Stx5ΔL lines (Supplementary Fig. 5g–k). The difference in PNA labeling between the HeLa Stx5ΔL cell lines therefore likely is attributable to clonal variation and/or off-target effects of CRISPR/Cas9. These results demonstrate that while the loss of Stx5S or Stx5L both result in *N*-glycosylation defects, the defect is stronger upon the loss of Stx5S, as this results in lower levels and a punctuated distribution of sialic acid moieties. Moreover, the mucin-type *O*-glycosylation defect seems specific to the loss of Stx5S. As these results reiterate the glycosylation defect observed on serum transferrin, total plasma *N*-glycans and apoCIII mucin *O*-glycans, patient fibroblasts are a suitable model to investigate the cell biological consequences of the complete disruption of the Stx5S isoform.

**Stx5M55V mutation results in mislocalization of glycosyltransferases.** Given that Stx5 mediates ER–Golgi trafficking[5,7,9–13,15,24,31], we next investigated whether the glycosylation defect in Stx5M55V patient fibroblasts was caused by the mislocalization of glycosyltransferases. We performed immunofluorescence labeling of mannosyl (α−1,3)-glycoprotein β-1,2-*N*-acetylglucosaminyltransferase (MGAT1, also known as GnTI), which catalyzes the addition of GlcNAc to the immature man-5 *N*-glycan. Compared to healthy donor fibroblasts, MGAT1 colocalizes only slightly less with the *cis*-Golgi marker GM130 in patient fibroblasts (Fig. 3a, b), but colocalized substantially less with the *trans*-Golgi network marker TGN46 (Fig. 3c, d). In addition, alpha-mannosidase 2 (MAN2A1), which catalyzes the final hydrolytic step in the *N*-glycan maturation pathway after

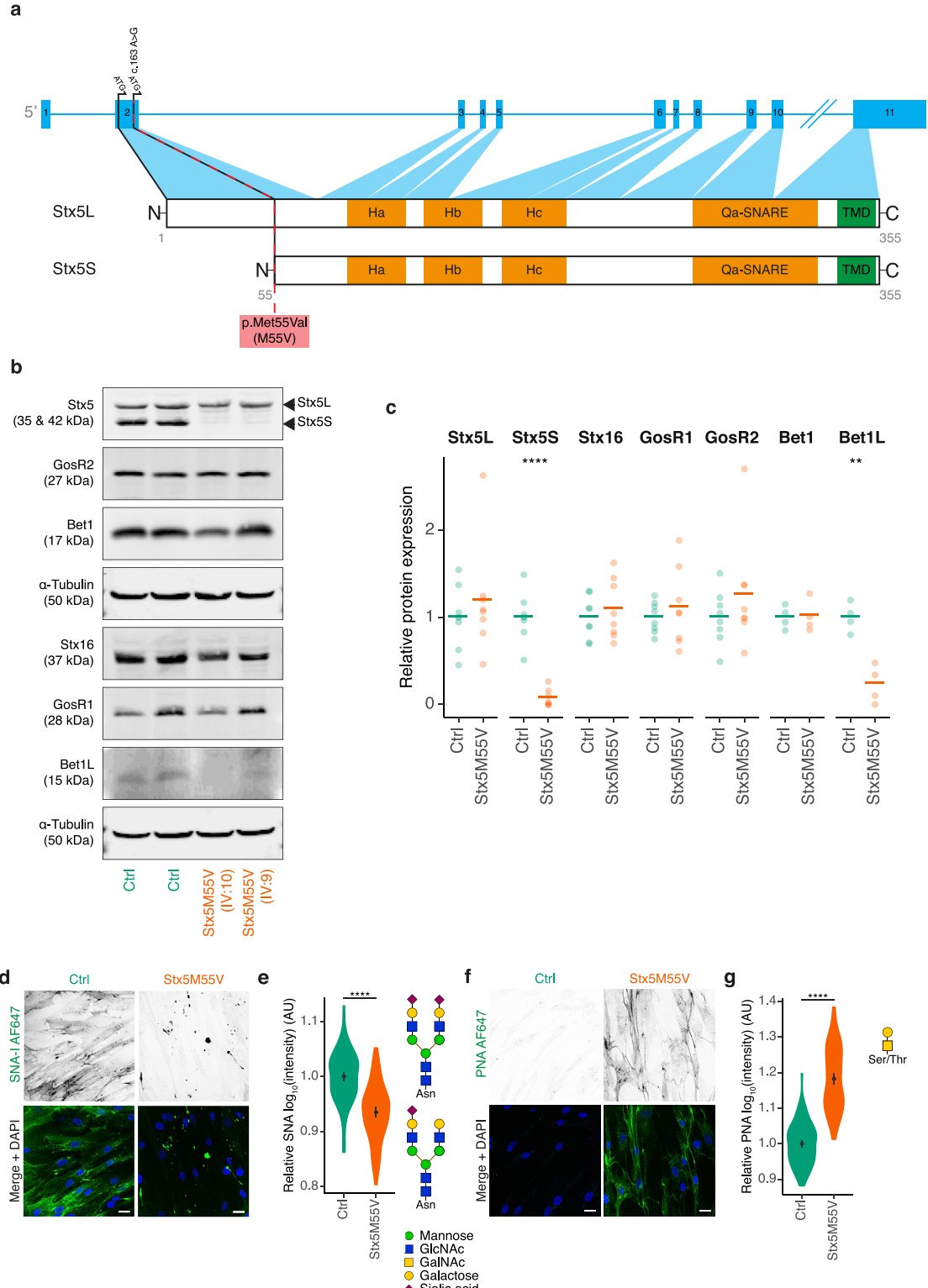

MGAT1 conversion, colocalized substantially less with both GM130 (Supplementary Fig. 6a, b) and TGN46 in patient fibroblasts (Supplementary Fig. 6c, d). Similarly to MGAT1, beta-galactoside alpha-2,6-sialyltransferase 1 (ST6GAL1), which catalyzes the transfer of sialic acid to galactose residues of *N*-glycans in an α-2,6 linkage, colocalized less with both GM130 (Supplementary Fig. 6e, f) and TGN46 in patient fibroblasts (Supplementary Fig. 6g, h). Finally, *N*-acetylgalactosaminyl-transferase 2 (GALNT2), which catalyzes the initial reaction in mucin *O*-linked glycan synthesis, localized more to the *cis*-Golgi (marker zinc finger protein-like 1 (ZFPL1)[34]) (Fig. 3e, f) and less to the *trans*-Golgi in patient fibroblasts (Fig. 3g, h).

To exclude the possibility that our conclusions were affected by potential mislocalization of TGN46 in the patient cells, we repeated these experiments but now co-stained for another *trans*-Golgi marker, the *trans*-Golgi coiled-coil protein p230[35] (Supplementary Fig. 7). Indeed, we observed similar Pearson correlation coefficients for p230 as for TGN46 with MGAT1,

**Fig. 2 Primary dermal fibroblasts are an accurate model of the glycosylation defect observed in Stx5M55V patients. a** Schematic representation of the intron-exon structure of *STX5* and the encoded proteoforms resulting from the two starting codons in exon 2. The Stx5M55V genetic variant is indicated by a dashed red line. Orange regions have a secondary helical structure. TMD, transmembrane domain. Ha, Hb, Hc: regulatory Habc-domain. **b** Immunoblot for SNARE proteins of cell lysates of primary human dermal fibroblasts from healthy donors (green, Ctrl) or Stx5M55V patients (orange, Stx5M55V). Fibroblasts were obtained from two unique healthy individuals and two patients. Lysates were loaded adjacent on single western blots that were sequentially probed for the mentioned proteins α-Tubulin loading control. **c** Quantification of (**b**). Band intensities were first normalized to the loading control, then to the average expression of both control lines. Each data point represents one cell line from two unique individuals tested at least twice (average shown as bar). Unpaired two-sided Student's *t*-test; $^{****}P = 1.1 \times 10^{-5}$; $^{**}P = 0.0017$; other comparisons not significant. **d** Fibroblasts of healthy donors (green, Ctrl) or Stx5M55V patients (orange, Stx5M55V) were probed with SNA-I lectin (green in merge). Representative confocal micrographs. Scale bars, 25 µm. DAPI in blue. **e** Quantification of (**d**). All data were $\log_{10}$-transformed and then normalized to the healthy donor. AU: arbitrary units. $N = 124$ (Ctrl) and 111 (Stx5M55V) cells from two unique individuals tested twice. Mean ± 95% CI. Unpaired two-sided Student's *t*-test. $^{****}P < 2.2 \times 10^{-16}$. **f, g** Same as panels (**d, e**), but now for PNA lectin. $N = 117$ (Ctrl) and 122 (Stx5M55V) cells from two unique individuals tested twice. Scale bars, 25 µm. Mean ± 95% CI. Unpaired two-sided Student's *t*-test. $^{****}P < 2.2 \times 10^{-16}$.

GALNT2, MAN2A1, and ST6GAL1 (Supplementary Fig. 7). Moreover, to confirm that the observed changes in colocalization were not due to lower expression of glycosyltransferases, we performed immunoblotting for several glycosyltransferases and could not detect consistent differences between control and Stx5M55V fibroblasts, although we noticed substantial variation in protein expression levels of some of the glycosyltransferases between the samples (Supplementary Fig. 8), reflecting expression differences among individuals and/or fibroblast lines.

Taken together, the loss of Stx5S results in irregular localization of glycosyltransferases to the Golgi apparatus. An altered Golgi organization and mislocalization of glycosyltransferases can have a profound impact on glycosylation as shown by computational simulations[36]. We investigated the organization of the Golgi complex in Stx5M55V fibroblasts in more detail.

**Loss of Stx5S alters the morphology of the ER and Golgi.** Transmission electron microscopy showed that the Golgi was not fragmented in the Stx5M55V patient fibroblasts. However, we observed dilation of rough ER and Golgi cisternae in the Stx5M55V patient fibroblasts (Fig. 4a–c and Supplementary Fig. 9), similar to previously observed alteration in the ultra-structure of the Golgi in several COG defects[37–39]. Notwith-standing these large alterations in Golgi morphology, the polarized arrangement of Golgi apparatus cisternae was still present in Stx5M55V, as observed by immunofluorescence labeling of *cis*- and *trans*-Golgi markers (Supplementary Fig. 10). These results indicate that although Stx5L is sufficient to maintain normal Golgi apparatus cisterna polarization, Stx5S is required for physiological ER and Golgi ultrastructure and proper traf-ficking of glycosylation enzymes.

To address the role of the Stx5 isoforms in more detail, we studied the distribution of Stx5 isoforms in the Golgi network. Because Stx5L contains an RKR ER-retrieval motif in its N-terminal extension, it locates more at the ER compared to Stx5S[13,18,20,21]. In line with this, we observed a more dominant localization of Stx5L at the ER and less at various Golgi compartments in Stx5M55V fibroblasts compared to total Stx5 localization in healthy control fibroblasts (Supplementary Figs. 11a, b, d, e and 12a, b, d, e).

A notable difference was the far more diffuse staining in Stx5M55V patients of the COPI coat protein βCOP (Supplemen-tary Fig. 11a, c) and of TGN46 (Supplementary Fig. 11d, f). In contrast, we observed a small increase in GM130 fluorescence in Stx5M55V patients (Supplementary Fig. 12f). Since western blot showed that total cellular levels of βCOP and GM130 were not consistently altered in Stx5M55V patients (Supplementary Fig. 11g), the more diffuse staining of βCOP is suggestive of reduced association with COPI-coated vesicles, while the higher staining intensity of GM130 suggests more association with the

*cis*-Golgi. In contrast, total TGN46 protein levels were somewhat reduced in patient fibroblasts (Supplementary Fig. 11h). These findings suggest that loss of Stx5S results in reduced COPI trafficking between GM130-marked *cis*- and TGN46-marked *trans*-Golgi compartments.

Using an antibody specific to Stx5L (antibody specificity validated in Supplementary Fig. 13), we next investigated whether the loss of Stx5S affects the intracellular distribution of Stx5L. An altered cellular localization of Stx5L might imply a compensatory mechanism. Indeed, in Stx5M55V, Stx5L localizes more to the *cis*-Golgi as measured by increased colocalization with GM130 (Fig. 4d, e). A slight increase in ER localization as measured by PDI colocalization was also observed (Fig. 4f, g). Taken together, the loss of Stx5S causes Stx5L to relocalize more towards the *cis*-Golgi. A possible explanation for the increased ER localization is that Stx5L will also recycle more to the ER due to its ER-retention signal[13,18,20–22], and is thereby displaced from other compartments such as trafficking intermediates and post-Golgi compartments.

**Loss of Stx5S compromises ER–Golgi trafficking.** As COPI is involved in retrograde Golgi–ER transport[40], we investigated whether trafficking at this interface is compromised in Stx5M55V fibroblasts by using the fungal metabolite brefeldin A (BFA), which inhibits COPI vesicle formation[41]. If loss of Stx5S results in reduced retrograde Golgi–ER transport, we expect reduced relo-calization of Golgi-resident proteins to ER upon BFA treatment. Indeed, redistribution of GALNT2 from the Golgi to the ER was incomplete in patient fibroblasts treated for 6 or 15 min with BFA (Fig. 5a–c and Supplementary Fig. 14), supporting a role for Stx5S in retrograde COPI trafficking. In addition, washout of BFA caused a more rapid localization of GALNT2 to the Golgi apparatus in Stx5M55V, likely due to its incomplete redistribu-tion to the ER (Fig. 5a–d and Supplementary Fig. 14a–c).

We investigated the role of Stx5L in Golgi trafficking further using temperature-synchronizable vesicular stomatitis virus G protein (VSVG) fused to GFP[42] in patient fibroblasts. At 40 °C, VSVG does not fold correctly and the VSVG-GFP protein remains trapped in the ER. A temperature shift to 32 °C enables the correct refolding of VSVG resulting in the synchronized release of VSVG-GFP from the ER, transit through the Golgi network, and finally delivery at the plasma membrane where it becomes accessible to antibody labeling. VSVG-EGFP appears at a Golgi-like compartment in both control and Stx5M55V fibroblasts at 30 min after temperature shift from 40 to 32 °C (Supplementary Fig. 14d), but after 60 min plasma membrane localization of VSVG is strongly reduced in Stx5M55V fibroblasts (Fig. 5e–g). Taken together, these findings show that Stx5L is sufficient for anterograde trafficking until the Golgi, but the loss of Stx5S significantly impairs intra- and/or post-Golgi trafficking, likely by the decrease of Stx5S-mediated intra-Golgi trafficking.

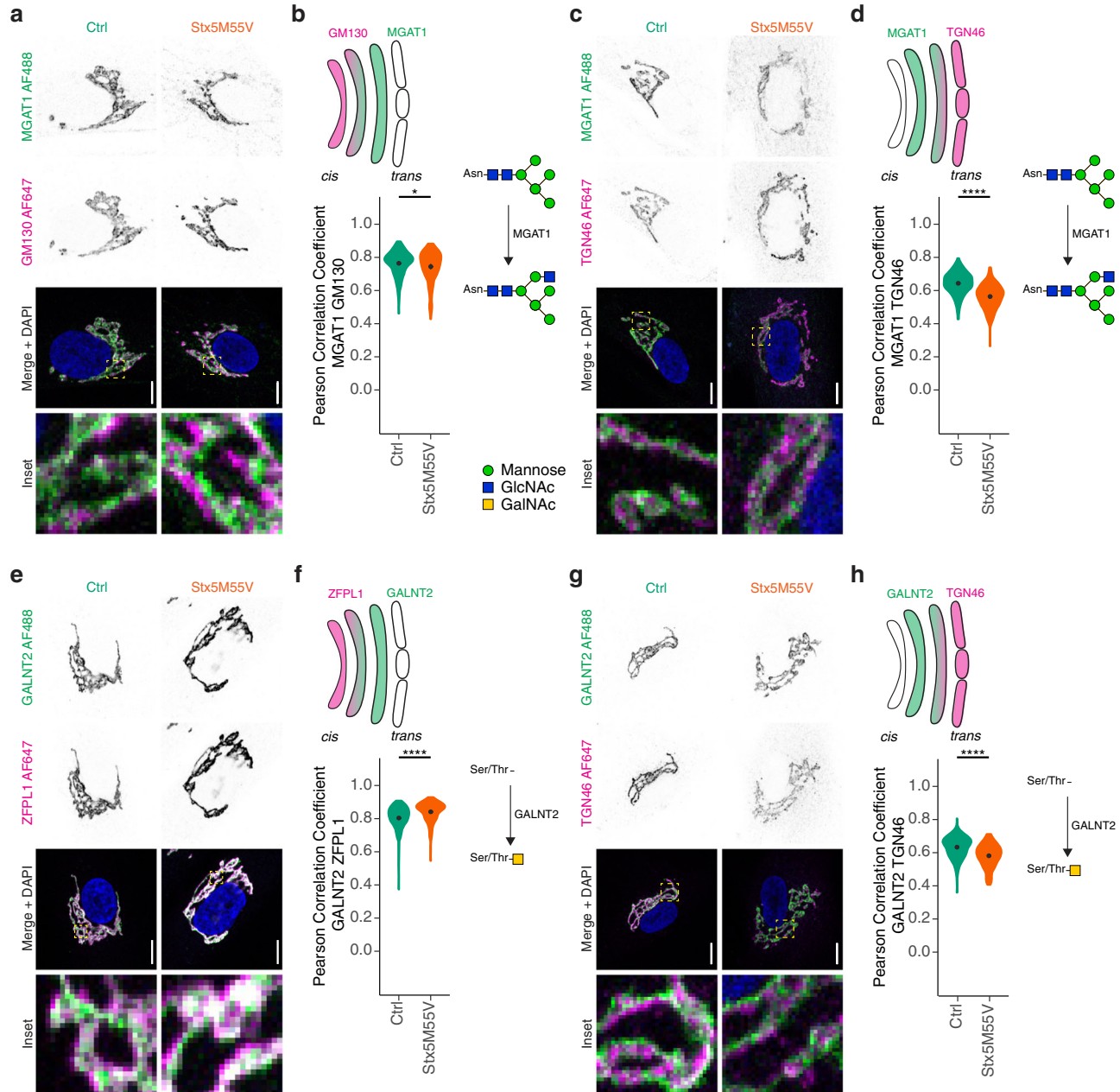

**Fig. 3 Glycosylation enzymes mislocalize in Stx5M55V patient fibroblasts. a** Immunofluorescence microscopy of MGAT1 (green in merge) and GM130 (magenta) in primary dermal fibroblasts of healthy donors (green, Ctrl) or Stx5M55V patients (orange, Stx5M55V). Representative confocal micrographs. Scale bars, 10 μm. DAPI in blue. $N = 157$ (Ctrl) and 126 (Stx5M55V) cells from two unique individuals tested twice. **b** Pearson's correlations coefficients between MGAT1 and GM130 of panel (**a**). $N = 157$ (Ctrl) and 126 (Stx5M55V) from two unique individuals tested twice. Mean ± 95% CI. Unpaired two-sided Student's $t$-test. $^{*}P = 0.047$. **c, d** Same as panels (**a, b**), but now for MGAT1 (green) and TGN46 (magenta). $N = 157$ (Ctrl) and 162 (Stx5M55V) cells from two unique individuals tested twice. Scale bars, 10 μm. Mean ± 95% CI. Unpaired two-sided Student's $t$-test. $^{****}P < 2.2 \times 10^{-16}$. **e, f** Same as panels (**a, b**), but now for GALNT2 (green) and ZFPL1 (magenta). $N = 240$ (Ctrl) and 146 (Stx5M55V) cells from two unique individuals tested twice. Scale bars, 10 μm. Mean ± 95% CI. Unpaired two-sided Student's $t$-test. $^{****}P = 5.9 \times 10^{-7}$. **g, h** Same as panels (**a, b**), but now for GALNT2 (green) and TGN46 (magenta). $N = 172$ (Ctrl) and 152 (Stx5M55V) cells from two unique individuals tested twice. Scale bars, 10 μm. Mean ± 95% CI. Unpaired two-sided Student's $t$-test. $^{****}P = 1.7 \times 10^{-11}$.

To further delineate the role of Stx5L in retrograde Golgi–ER trafficking, we performed a BFA experiment in the Stx5L-lacking HeLa cells. In these cells, BFA resulted in faster relocalization of GALNT2 to the ER compared to parental HeLa (Supplementary Fig. 15a, b), indicating that Stx5S suffices for retrograde COPI trafficking and the expression of Stx5L counteracts this process. Further investigation of anterograde ER–Golgi trafficking in Stx5ΔL cells with H-89 washout (Supplementary Fig. 15c, d), the

retention using selective hooks (RUSH) system for synchronized ER–Golgi transport[43] (Supplementary Fig. 15e, f, Supplementary Movies 1, 2), and temperature-synchronizable VSVG[42] (Supplementary Fig. 15g, h, Supplementary Movies 3, 4) revealed no phenotype relating to the loss of Stx5L. Thus, these data suggest Stx5L has no necessary function in ER–Golgi trafficking as Stx5S can compensate, while Stx5L can only partly compensate for the loss of Stx5S in retrograde Golgi–ER and intra-Golgi transport.

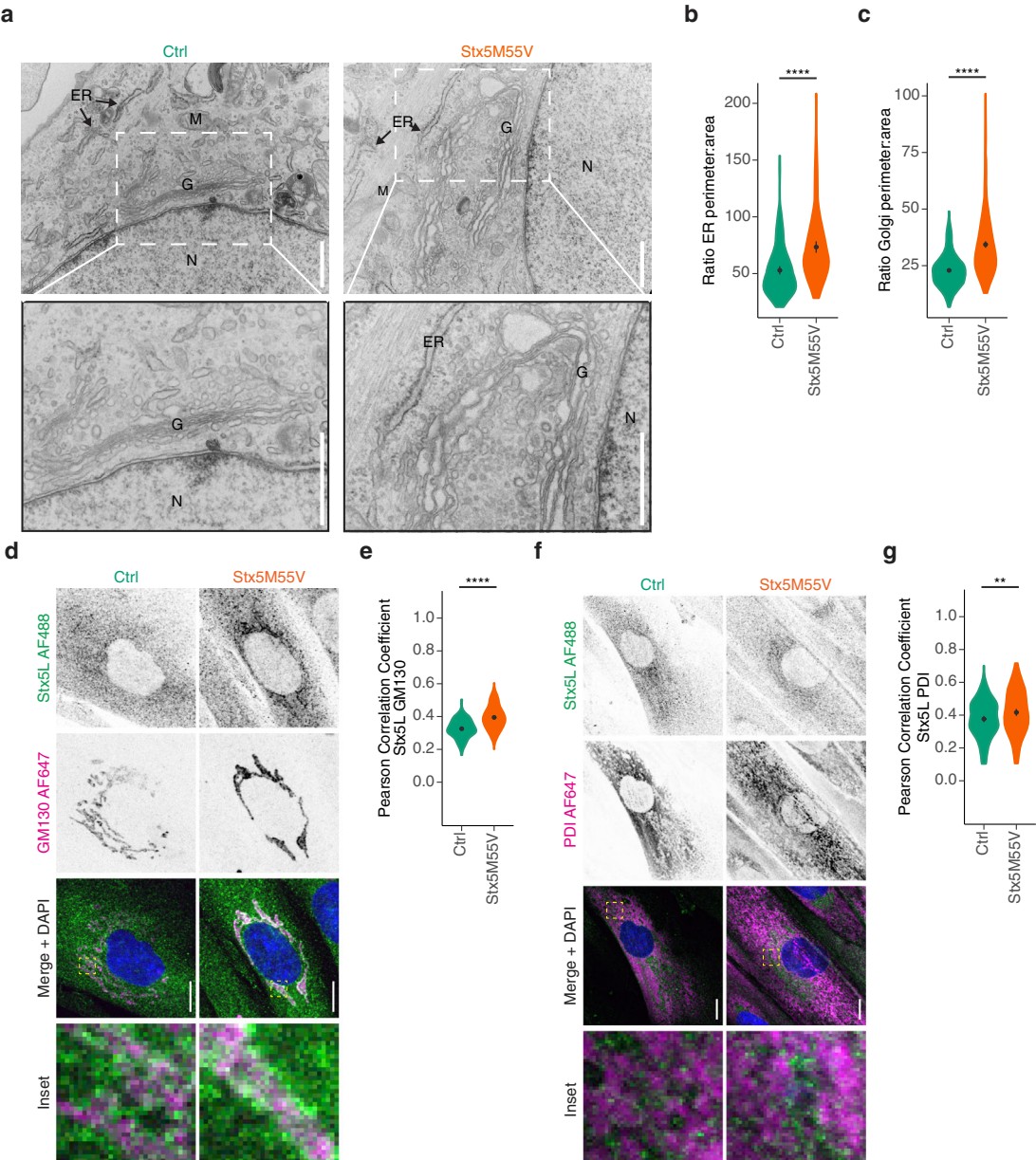

**Fig. 4 Loss of Stx5S alters ER and Golgi morphology. a** Representative transmission electron micrographs from healthy donor fibroblasts (left) or Stx5M55V patient fibroblasts (right). Scale bars, 1 μm. N, nucleus; G, Golgi apparatus; ER, endoplasmic reticulum; M, mitochondrion. More electron micrographs in Supplementary Fig. 9. **b** ER perimeter and area quantification of panel (**a**), the ratios of the perimeters over the areas are plotted. $N = 144$ (both Ctrl and Stx5M55V) ER sections. Mean ± 95% CI. Unpaired two-sided Student's $t$-test. $^{****}P = 3.8 \times 10^{-10}$. **c** Same as panel (**b**), but now for Golgi. $N = 400$ (both Ctrl and Stx5M55V) Golgi sections. Mean ± 95% CI. Unpaired two-sided Student's $t$-test. $^{****}P < 2.2 \times 10^{-16}$. **d** Immunofluorescence microscopy of Stx5L (green in merge) and GM130 (magenta) in primary dermal fibroblasts of healthy donors (green, Ctrl) or Stx5M55V patients (orange, Stx5M55V). Representative confocal micrographs. Scale bars, 10 μm. DAPI in blue. **e** Pearson's correlations coefficients between Stx5L and GM130 of panel (**d**). $N = 102$ (Ctrl) and 135 (Stx5M55V) cells from 2 unique individuals tested twice. Mean ± 95CI. Unpaired two-sided Student's $t$-test. $^{****}P = 9.45 \times 10^{-14}$. **f, g** Same as panels (**d, e**), but now for Stx5L (green) and TGN46 (magenta). $N = 147$ (Ctrl) and 156 (Stx5M55V) from 2 unique individuals tested twice. Scale bars, 10 μm. Mean ± 95CI. Unpaired two-sided Student's $t$-test. $^{**}P = 0.0051$.

**The two isoforms of Stx5 differently engage in SNARE complexes.** Our results in patient fibroblasts indicate differential trafficking roles of the two Stx5 isoforms in ER–Golgi trafficking. One possible explanation for this is that Stx5S and Stx5L have different trafficking rates between the ER and the Golgi. To test this, we fused each Stx5 isoform to streptavidin-binding protein (SBP) and mCitrine (Stx5L-SBP-mCitrine and Stx5S-SBP-mCitrine; Stx5L-SBP-mCitrine carries the M55V mutation to suppress the expression of Stx5S). Moreover, we generated a mutant form of Stx5L where the RKR ER-retrieval motif was converted to 3x alanine (AAA) (Stx5LΔER-SBP-

mCitrine)[18], to delineate the role of this motif in ER–Golgi transport. The co-expression of these constructs with ER-localized streptavidin enabled the synchronized release of the Stx5 fusion proteins from the ER using biotin, which is the so-called RUSH system[43] (Supplementary Fig. 16a). Co-expressing each Stx5 isoform with the Golgi marker Giantin fused to mScarlet[44] in HeLa cells, allowed to visualize the trafficking of Stx5-SBP-mCitrine to the Golgi following the addition of biotin (Supplementary Fig. 16a, b and Supplementary Movies 5–7). All three constructs were expressed at similar levels, as judged from the fluorescent intensities. However, all Stx5 forms

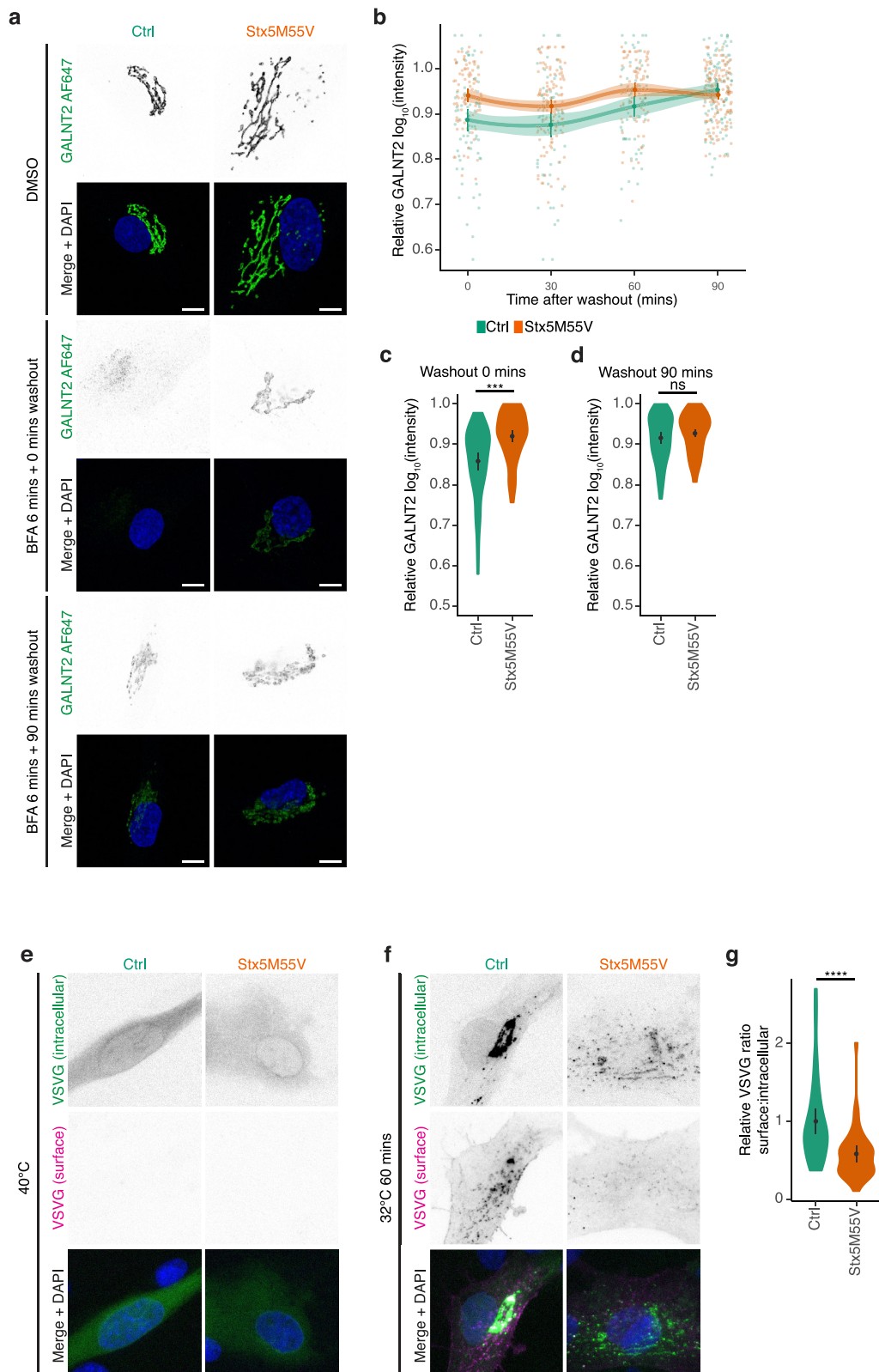

reached the Golgi with the same rate and achieved maximal Golgi localization after about 20 min (Supplementary Fig. 16c, d). We also did not observe any differences in the exit of the two isoforms from the Golgi (Supplementary Fig. 16c, e). This suggests that the two isoforms traffic with similar rates between the ER and Golgi and that this is not dependent on the RKR ER-retrieval motif of Stx5L. The altered morphologies of ER and Golgi, and the compromised

ER–Golgi trafficking are thus not caused by different trafficking rates of the two Stx5 isoforms.

Another explanation for the different trafficking roles of the two Stx5 isoforms is that they differently engage in SNARE complexes. Since interactions of Stx5 with Bet1 and Bet1L mediate anterograde ER–Golgi transport and retrograde intra-Golgi transport, respectively[2–5], we hypothesized that Stx5S

**Fig. 5 Loss of Stx5S compromises ER–Golgi trafficking. a** Immunofluorescence microscopy of GALNT2 (green in merge) in primary human dermal fibroblasts of healthy donors (green, Ctrl) or Stx5M55V patients (orange, Stx5M55V) in the absence or presence of brefeldin A (BFA) for 6 min and washout for the indicated times. Representative confocal micrographs are shown. Scale bars, 10 μm. DAPI in blue. **b** Time course of relative maximum fluorescence intensities of GALNT2 from panel (**a**). All data were normalized to the DMSO condition (vehicle). Mean ± 95% CI. Error bands are extrapolated 95% CI between the timepoints. **c** Quantification of the 0 min washout timepoint from panels (**a**, **b**). $N = 76$ (Ctrl) and 85 (Stx5M55V) cells from two unique individuals tested twice. Mean ± 95CI. Unpaired two-sided Student's *t*-test, followed by Bonferroni correction for multiple comparisons. ***$P = 0.00080$. **d** Same as (**c**), but now for the 90 min timepoint. $N = 97$ (Ctrl) and 152 (Stx5M55V) cells from two unique individuals tested twice. Mean ± 95% CI. Unpaired two-sided Student's *t*-test, followed by Bonferroni correction for multiple comparisons; not significant. **e** Immunofluorescence microscopy of VSVG-EGFP in primary human dermal fibroblasts of healthy donors (green, Ctrl) or Stx5M55V patients (orange, Stx5M55V) cultured overnight at 40 °C. Representative confocal micrographs are shown. Scale bars, 10 μm. DAPI in blue. **f** Same as (**e**), but now after 60 min at 32 °C. Scale bars, 10 μm. **g** Quantification of the ratio of surface to intracellular of VSVG after 60 min at 32 °C. $N = 45$ (Ctrl) and 39 (Stx5M55V) cells from two unique individuals tested twice. Mean ± 95% CI. Unpaired two-sided Student's *t*-test. ****$P = 4.3 \times 10^{-5}$.

would interact more strongly with Bet1L. To test this, we developed an approach to visualize SNARE complexes based on a combination of the RUSH system[43] and our previously developed FRET-FLIM approach for visualization of SNARE complexes[45] (Fig. 6a). This FRET-FLIM approach employed Stx5 isoforms C-terminally fused with a donor fluorophore (mCitrine) and Bet1L C-terminally fused with an acceptor fluorophore (mCherry). The formation of a post-fusion SNARE complex results in the close proximity of the donor and an acceptor fluorophore resulting in FRET, which can be measured from a decreased donor fluorescence lifetime ($\tau$). Contrary to ratiometric FRET, FRET-FLIM is not dependent on local concentration differences or excitation intensities of the donor and acceptor fluorophores, as $\tau$ is an intrinsic property of the fluorophore itself. By combining the FRET-FLIM approach with the RUSH system, we were able to control the spatial localization of Stx5 isoforms and measure interactions specifically at the ER (no biotin) or the Golgi apparatus (20 min after biotin addition; Fig. 6b, c). At 30 min prior to imaging, cells were incubated with cycloheximide in culture medium to make sure background interaction from any ER-localized newly synthesized acceptor construct was mitigated.

For the mCitrine donor-only Stx5 constructs, we measured similar apparent fluorescence lifetimes for both isoforms (Fig. 6c and Supplementary Fig. 17a, Stx5L: 2.92 ± 0.02 ns, Stx5S: 2.92 ± 0.02 ns (mean ± 95% CI)) prior to biotin addition, while these lifetimes slightly decreased following biotin addition (Fig. 6d and Supplementary Fig. 17a, Stx5L: 2.86 ± 0.03 ns, Stx5S: 2.85 ± 0.02 ns (mean ± 95% CI)). We attribute this reduced lifetime to the fact that mCitrine is somewhat pH-sensitive[46] and the pH of the Golgi apparatus is lower than in the ER lumen[47]. We then co-expressed the Stx5 isoforms with mCherry-tagged Bet1L (Bet1L-mCherry) (Fig. 6a, b). At the ER, thus before the release of Stx5 with biotin, we observed reduced average apparent fluorescence lifetimes for both Stx5S and Stx5L with Bet1L-mCherry, compared to the donor-only controls (Fig. 6b, c, Stx5L: 2.48 ± 0.07 ns, Stx5S: 2.49 ± 0.06 ns (mean ± 95% CI)), whereas the lifetimes of Stx5S and Stx5L did not significantly differ from each other. After the release in the presence of biotin, this difference between Stx5L and Stx5S became significant and average apparent fluorescence lifetimes were 2.33 ns (±0.06 95% CI) for Stx5L while Stx5S dropped to 2.24 ns (±0.07 95% CI) (Fig. 6b, d). To validate that the observed effect is indeed caused by functional SNARE complex formation, we repeated this experiment with VAMP8 instead of Bet1L as the acceptor R-SNARE. VAMP8 has no role in ER–Golgi membrane fusion but rather associates with the late endosomal/lysosomal compartment[33,45,48–52]. We only observed minor decreases in average apparent fluorescence lifetimes for both Stx5L and Stx5S (Supplementary Fig. 17b, c, prior to biotin Stx5L: 2.83 ± 0.03 ns, Stx5S: 2.82 ± 0.03 ns (mean ± 95% CI), upon biotin addition Supplementary Fig. 17b, d, Stx5L: 2.78 ± 0.04 ns, Stx5S:

2.76 ± 0.03 ns (mean ± 95% CI)). These FLIM results demonstrate that Stx5S interacts more strongly with Bet1L at the Golgi than Stx5L. Thus, Stx5S is the dominant Qa-SNARE for intra-Golgi trafficking.

## Discussion

Since the advent of the genomic age, close to 6000 monogenic disorders have been discovered[53]. While nearly all of these disorders result in a truncated, unstable and/or nonfunctional protein, e.g., due to a genetic variant in the catalytic site or protein misfolding, isoform-specific mutations are rare. Here we identified a mutation in an alternate site of ribosomal translation leading to human disease, namely the mutation of the second starting methionine of Stx5. This mutation leads to the complete and specific loss of Stx5S. Although *STX5* is an essential gene for embryonic development in mice[16,17], here we show that in humans the loss of Stx5S still allowed a completed pregnancy. Nevertheless, patients have a very severe clinical pathology characterized by infantile mortality due to liver disease, skeletal abnormalities, and protein glycosylation defects. While the exact mechanism for alternative translation is unclear, this might be an actively regulated process. It could also be simply regulated by the affinity of the ribosome for the nucleotide sequence upstream of the starting codon. Supporting the latter option, analysis of translation initiation sites with NetStart[54] revealed that the starting codon of Stx5S is located in a more optimal nucleotide context than the starting codon for Stx5L (Supplementary Fig. 18). This could lead to more leaky ribosomal scanning[55,56], resulting in the more or less equimolar ratio of the expression of Stx5L and Stx5S that we observed in fibroblasts and PBMCs of healthy individuals. On the other hand, western blot revealed different ratios of the two Stx5 isoforms in different organs in rats[18], suggesting (i) that different cell types express different levels of Stx5S and Stx5L and this is probably related to their exocytic function, and (ii) that the initiation of starting translation might be regulated and not merely dependent on the binding affinities of the ribosome.

Co-fractionation and microscopy studies have revealed that the localization of Stx5L and Stx5S overlap to a large extent, but that they are generally distributed as a gradient between ER, ERGIC, and Golgi apparatus[20]. This observation has previously led to the suggestion that Stx5L might play a role in early Golgi trafficking, while Stx5S functions in late Golgi trafficking[5,7,9–13,15,24,31]. Our data now show that this is not the case and that both Stx5 isoforms can mediate both early and late anterograde and retrograde Golgi trafficking with sufficient fidelity to keep the layered Golgi morphology intact. However, the role of Stx5S is more important for retrograde Golgi–ER and intra-Golgi trafficking, and its absence leads to an altered ER and Golgi morphology, and altered distributions of glycosylation enzymes and trafficking proteins. The cumulative effect of slight mislocalization of all

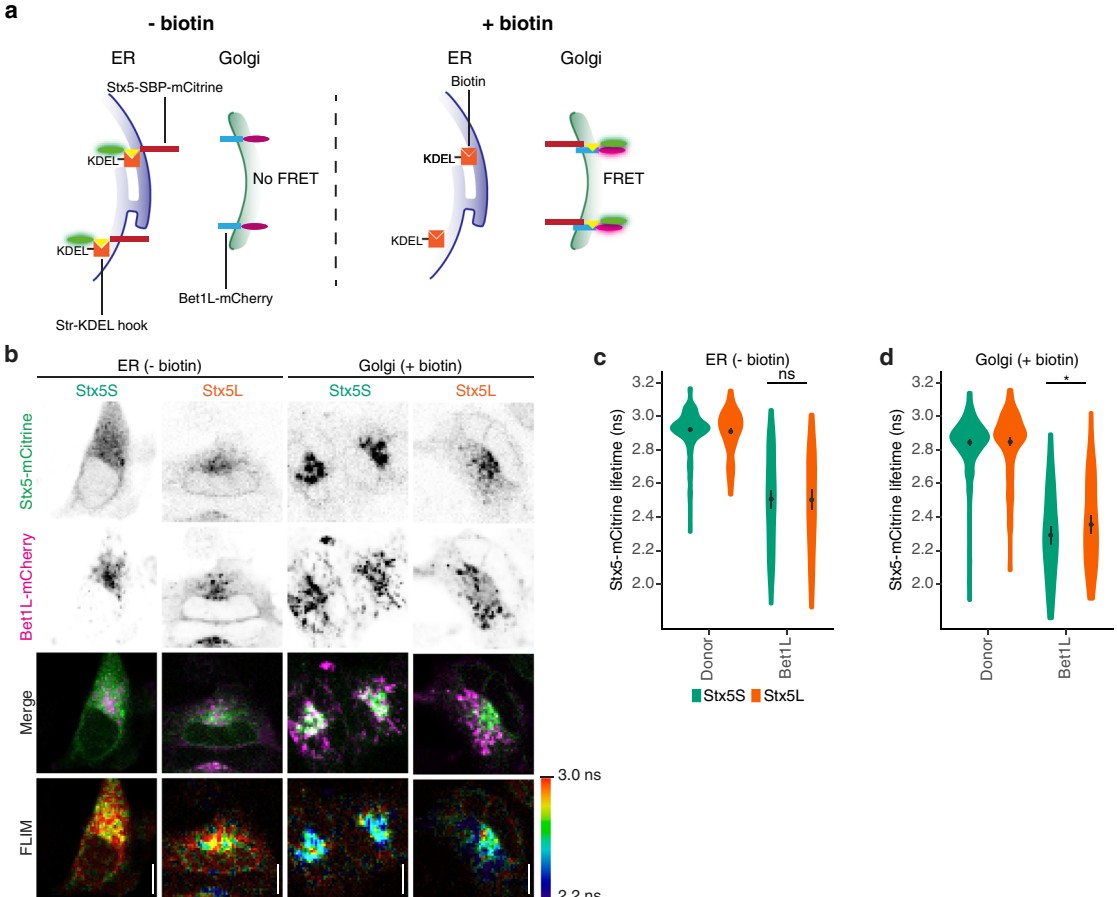

**Fig. 6 Stx5S is the dominant Qa-SNARE for intra-Golgi trafficking. a** Schematic overview of experimental design for complex formation of Stx5 isoforms with Bet1L, based on the RUSH system and SNARE complex measurement by FRET-FLIM. In the absence of biotin (left panel), the reporter cargo (Stx5-SBP-mCitrine) is trapped at the ER by the lumenal Str-KDEL hook, and no FRET with Golgi-localized Bet1L-mCherry occurs. When biotin is added (right panel), biotin outcompetes the interaction with streptavidin, allowing Stx5-SBP-mCitrine to traffic freely to its destination compartment, and SNARE complex formation with Golgi-localized Bet1L-mCherry results in FRET. SBP, streptavidin-binding protein; Str, streptavidin; FRET, Förster resonant energy transfer. FLIM, fluorescence lifetime imaging microscopy. **b** Representative confocal micrographs and FLIM images of HeLa cells co-expressing Stx5-mCitrine (green in merge) and Bet1L-mCherry (magenta) without (ER) or with (Golgi) biotin. Scale bars, 10 μm. **c**, **d** Stx5-mCitrine lifetimes at the ER (**c**) and Golgi (**d**) from panel (**b**). Donor: donor-only control with cells only expressing Stx5S-SBP-mCitrine or Stx5L-SBP-mCitrine. $N = 210$ (Stx5S Donor ER), 202 (Stx5L Donor ER), 93 (Stx5S Bet1L ER), 83 (Stx5L Bet1L ER), 228 (Stx5S Donor Golgi), 228 (Stx5L Donor Golgi), 88 (Stx5S Bet1L Golgi), and 88 (Stx5L Bet1L Golgi) cells from four to six independent experiments. Mean ± 95% CI. Paired two-sided Student's $t$-test after taking the average for each experiment. ns: not significant; $^*P = 0.012$. Donor-only controls are not significant.

glycosyltransferases might well explain the observed hypoglycosylation in the Stx5M55V patients. Indeed, recent modeling showed that the slight mislocalization of glycosyltransferases can result in large differences in glycosylation patterns, because glycosylation is the result of the sequential addition and removal of different sugar moieties at various Golgi compartments[36].

The dominant role of Stx5S in retrograde Golgi–ER and intra-Golgi trafficking is corroborated by the observation that cellular levels of Bet1L, with known roles in intra-Golgi trafficking, are lower in Stx5M55V patient cells. Interestingly, genetic variants in COG tethering complex components, which are also implicated in CDGs, also resulted in lower levels of Bet1L as well, and this was attributed to a mislocalization of Bet1L to the ER where it was degraded[57]. Possibly, a similar mechanism might explain the reduction in Bet1L levels in Stx5M55V patients. Although the Stx5-Bet1L interaction has been reported in several studies[14,40], our study now shows this interaction in situ using FLIM. This interaction is localization dependent and occurs mostly when Stx5 is localized at the Golgi. Moreover, we observed stronger interaction of Bet1L with Stx5S compared to Stx5L at the Golgi,

which is likely the result of the differential localization of both isoforms.

An important function of the Golgi apparatus is protein glycosylation[58]. Collectively, somatic mutations affecting glycosylation are classified as CDGs and currently over 100 monogenic diseases affecting glycosylation have been identified[59,60]. A significant number of these include defects in Golgi trafficking, such as the components of the COG tethering complex[37,61–68], mutations in genes coding for the vacuolar H$^+$-ATPase and its assembly factors[69–72], and novel genes involved in Golgi ion homeostasis[73–75]. Furthermore, defects are known in components associated with COPI-coated vesicles[76] that result in deficient protein glycosylation in patient cells, but are not linked to abnormal glycosylation of proteins in plasma and thus escape routine CDG screening. Our study revealed that mutations in ER–Golgi SNAREs can also lead to CDG, thus highlighting the potential of glycosylation screening in patients to uncover novel cell biological mechanisms.

While the cellular effects of the loss of Stx5S in Stx5M55V mutant fibroblasts are subtle, there can be pronounced

consequences in secretory cells, such as exocrine and endocrine cells, which are sensitive to minor disruptions of the secretory pathway[58,76–78]. The defects in intra-Golgi trafficking can explain the hypoglycosylation not only in Stx5M55V patients, but also of other pathologies. For instance, Stx5 can participate in the trafficking and processing of the very-low-density lipoprotein receptor (VLDL-R) and this role is dependent on the expression of Stx5[79], thus providing an explanation for the observed cholesterol homeostasis defect with elevated cholesterol in all Stx5M55V patients.

In summary, we have demonstrated that a mutation in an alternative translation start site in Stx5 affects intracellular membrane trafficking, leading to CDG. While there are previous descriptions of alternative start site variations leading to disease[80], this study shows that mutations in an alternative starting codon causing the loss of an isoform can also lead to disease.

## Methods

**Ethics**. The study was approved by Research Ethics Committee of the University of Tartu (approval dates 19.12.2011, 20.02.2012, and 17.03.2014, and approval numbers 210/M-17, 212/M-31 and 235/M-13, and 17.03.2014, respectively) and were strictly in accordance with the Declaration of Helsinki. Informed consent for carrying out research was obtained from the family of investigated individuals. Buffy coats and whole blood were obtained as anonymous coded specimens from the Dutch blood bank (Sanquin) and were handled according to known practice and legal guidelines. The research with human blood samples at the Department of Tumor Immunology complies with all institutional and national ethics regulations and has been approved by the ethics committee of Sanquin.

**Glycosylation studies**. Screening for CDG by isoelectric focusing (IEF) was carried out as before[70] as follows: for N-glycosylation screening, transferrin IEF was performed. Then, 10 μL serum or plasma was added to a solution containing iron and NaHCO₃, electrophorized on a 5–7 pH gradient gel, and incubated with 60 μL polyclonal rabbit anti-transferrin antibody (Dako, #A0061). Quantification of the gel was done with Image Quant software (TotalLab), by expression of the intensity of the individual transferrin glycoforms as relative percentage of the total intensity of transferrin bands. For mucin O-glycosylation screen, apolipoprotein CIII (apoCIII) IEF was performed. Then, 2 μL of serum or plasma was 15× diluted with saline solution. Before electrophoresis, the gel was rehydrated in a solution containing 8 M urea. After blotting on a nitrocellulose membrane filter, the blot was washed and blocked before incubation with anti-ApoCIII (1:2000, Rockland, #600-101-114). After incubation with the secondary anti-goat-HRP antibody (1:5000, Thermo Scientific, #31402) and ECL reagent (Pierce), the blot was visualized on a LAS3000 imaging system (Fujifilm). Blot quantification was done with Image Quant software by expression of the intensity of the individual apoCIII glycoforms as relative percent of the total intensity of apoCIII bands.

Plasma N-glycan profiling was performed by MALDI-TOF mass spectrometry of permethylated glycans[81], using 10 μL plasma. Glycans from 10 μL serum or plasma were cleaved with PNGaseF (NE Biolabs) and incubated overnight. After purification on graphitized carbon SPE columns, the glycans were permethylated, purified again, and eluted in 50 μL of 75% v/v aqueous acetonitrile. The glycans were dried and resuspended in a methanol/sodium acetate mixture for spotting. Measurements were done on a AB SCIEX 5800 MALDI TOF/TOF (Sciex). High-resolution mass spectrometry of intact transferrin was performed on a 6540 nanochip QTOF (Agilent), according to published protocols[82] as follows: 10 μL of serum or plasma sample was incubated with anti-Tf beads before injection, and the eluate was analyzed on a microfluidic nanoLC-C8-chip 6540 QTOF instrument (Agilent technologies). Agilent Mass Hunter Qualitative Analysis Software B.04.00 was used for data analysis. For deconvolution of the charge distribution raw data, Agilent BioConfirm Software was used.

**Microarray analysis**. DNA was extracted either from peripheral blood according to the standard salting out protocol (IV:9 and IV:10) or from amnionic fluid cell culture (IV:8). Screening for chromosomal abnormalities was performed using HumanCytoSNP-12 BeadChips (Illumina Inc., San Diego, CA, USA). In all, 200 ng of total DNA per sample was processed according to the protocol supplied by the manufacturer. Genotypes were called by GenomeStudio v2011.1 software and the data were analyzed using GenomeStudio Genome Viewer tool (Illumina Inc.). The minimum threshold for LCSH regions was set at 5 Mb.

**Exome sequencing**. Genomic DNA was extracted from fibroblasts from patient IV:9 according to the manufacturer's protocol using a Qiagen Mini Kit (Qiagen) and was checked for DNA degradation on agarose gels. Next-generation sequencing (NGS) and analysis were performed as described[71] previously, as follows: exome enrichment was performed using the SureSelect Human All Exon 50 Mb Kit

(Agilent), covering ~21,000 genes. The exome library was sequenced on a SOLiD 5500xl sequencer (Life Technologies). Color space reads were iteratively mapped to the hg19 reference genome with the SOLiD LifeScope software version 2.1. Called variants and indels were annotated using an in-house annotation pipeline[83,84] and common variants were filtered out based on a frequency of >0.5% in dbSNP and a frequency of >0.3% in our in-house database of >5000 exomes. Quality criteria were applied to filter out variants with less than 5 variant reads and less than 20% variation. Furthermore, synonymous variants, deep intronic, intergenic, and UTR variants were excluded. The identified variant was confirmed by Sanger sequencing in all the affected individuals (IV:8, IV:9, and IV:10) and their mother (III:2). Paternal DNA (III:1) was not available.

**Cell culture**. HeLa cells (authenticated by ATCC through their human STR profiling cell authentication service), including Stx5ΔL cell lines, were maintained in high-glucose DMEM with Glutamax (Gibco 31966021). Human primary dermal fibroblasts were obtained from patients or healthy donors and maintained in Medium 199 with EBSS and L-glutamine (Lonza BE12-119F). All media were supplemented with 10% fetal calf serum (FCS, Greiner Bio-one, Kremsmünster, Austria) and antibiotic-antimycotic solution (Gibco 15240-062). All cells were regularly tested for mycoplasma contamination. Human PBMCs were obtained from buffy coats by Ficoll density centrifugation[85].

**Plasmids and transfection**. Str-KDEL_ManII-SBP-EGFP was a gift from Franck Perez (Addgene plasmid #65252). VAMP8-mCherry was constructed earlier[45] and previously deposited to Addgene (Addgene plasmid #92424). Str-KDEL_Stx5L-SBP-mCitrine and Str-KDEL_Stx5S-SBP-mCitrine were constructed by replacing the ManII-SBP-EGFP cassette in Str-KDEL_ManII-SBP-EGFP using the AscI and XbaI restriction sites. Stx5 coding sequences were codon-optimized for Homo sapiens using JCat and ordered from Genscript, the Stx5L coding sequence carries the M55V mutation to suppress the production of Stx5S. Stx5LΔER coding sequence was generated with Q5-polymerase site-directed mutagenesis, using the Stx5L cDNA as a template with the following primer: 5′- CTTCG AATTC AATGA TTCCG GCCGC CGCCT ACGGC AGCAA GAACA CC. Sequences were verified with Sanger sequencing. HeLa cells were transfected with plasmid vectors using Fugene HD (Promega E2311), using the recommended protocol of the manufacturer. Only cells expressing low to moderate levels of the transfected plasmids, based on fluorescence intensity and manual localization scoring, were chosen for subsequent microscopic analyses.

**CRISPR/Cas9**. Stable knock out of Stx5L in fibroblasts and HeLa cells was obtained using the CRISPR-CAS9 method. For this, pairs of gRNA sequences were designed upstream of the STX5 initiation codon (crispr.mit.edu, pair 1: ATAAC CTCGG ACTGT TGTGG AGG and ATGAT CCCGC GGAAA CGCTA CGG; pair 2: TAACC TCGGA CTGTT GTGGA GGG and TGATC CCGCG GAAAC GCTAC GGG). The gRNA sequences were cloned in pSpCas9n(BB)-2A-Puro (PX462) V2.0 (gift from Feng Zhang, Addgene no. 62987)[86] and transfected into fibroblasts or HeLa cells by electroporation (Neon Transfection System, Thermo-Fisher, MA, USA). After initial selection with puromycin, the medium was changed for conditioned medium (collected from parental fibroblasts or wild-type HeLa cells at 70% confluency) supplemented 1:1 with fresh medium. Single clones were obtained and screened for knockout of Stx5L by SDS-PAGE and western blotting.

**Immunofluorescence**. Cells were plated on cleaned 12 mm glass coverslips (Electron Microscopy Services, 72230-01) and on the following day fixed with 4% paraformaldehyde for 15 min at room temperature (RT). Following quenching with 50 mM NH₄Cl in PBS, cells were permeabilized and blocked in 2% normal donkey serum (Rockland, 017-000-121) and 0.1% saponin (permeabilization buffer) for 30 min at RT. Primary and secondary antibodies (list of antibodies and dilutions in Supplementary Table 6) were diluted in permeabilization buffer and incubated for 1 h at RT. Finally, cells were washed with 0.1% Triton X-100 in PBS to remove background staining and mounted with mounting medium containing 0.01% Trolox (6-hydroxy-2,5,7,8-tetramethylchroman-2-carboxylic acid) and 68% glycerol in 200 mM sodium phosphate buffer at pH 7.5 with 0.1 μg/mL DAPI. Coverslips were sealed with nail polish. Cells were imaged on a Leica SP8 SMD confocal laser scanning microscope, equipped with an HC PL APO CS2 63x/1.20 WATER objective. Colocalization analysis was performed using the *pearsonr* function from the Python package SciPy[87]. Individual cells were first saved to separate .tiff files with ImageJ without any modifications, and then processed in a fully automated and unbiased fashion using the *pearsonr* function.

**Lectin stainings**. Cells were plated on cleaned 12 mm glass coverslips for microscopy or 6-well plates for flow cytometry, and after 72 h culturing fixed with 4% paraformaldehyde. Cells were blocked with Carbo-Free Blocking Solution (Vector Laboratories, SP-5040) and incubated with 4 μg/mL biotinylated SNA-I (Vector Laboratories, B-1305) or PNA (Vector Laboratories, B-1075) diluted in Carbo-Free Blocking Solution. Cells were then incubated with Streptavidin-Alexa Fluor 647 (ThermoScientific, S32357) before coverslips were mounted as described above. Cells were imaged on a Leica SP8 SMD confocal laser scanning microscope, equipped with an HC PL APO CS2 63x/1.20 WATER objective. For flow

cytometry, cells were resuspended in FACS buffer (phosphate buffered saline + 0.5% FBS + 0.01% NaN₃). Flow cytometry samples were run on a FACSLyric flow cytometer (BD Biosciences) and analyzed with FlowJo X (FlowJo, LLC).

**Brefeldin A assay**. Fibroblasts were plated in black clear-bottom 96-well plates (Greiner, 655090) and cultured until 80% confluent. Cells were treated with either 10 μg/mL BFA in DMSO (Cayman Chemicals, 11861) or DMSO alone for 6 or 15 min in a humified condition. Washout was performed by washing five times with Leibovitz's L-15 (Gibco 21083027) with 10% FBS, then incubated in Leibovitz's L-15 with 10% FBS at 37 °C for the indicated times. After incubation, plates were transferred immediately to ice and cells were fixed with 4% paraformaldehyde, after which the above immunofluorescence protocol was performed. Microscopy images of the washout samples were acquired using a Leica SP8 SMD confocal laser scanning microscope, equipped with an HC PL APO CS2 63x/1.20 WATER objective. Microscopy images of the 15 min incubation samples were acquired using a Leica high-content microscopy system based on a Leica DMI6000B (Leica Microsystems) and an HCX PL S-APO 40.0 × 0.75 DRY objective. HeLa cells were plated on 12 mm coverslips and incubated in the same way with BFA, but fixed with 100% methanol at −20 °C for 15 min. Imaging of these samples was performed using a Leica DMI6000B epifluorescence microscope equipped with an HC PL APO 63 × 1.40 OIL objective. Cells were analyzed using Fiji (http://fiji.sc/) by first removing noise outliers (bright outliers, radius 2.0 pixels, threshold 50), then manually selecting cells and measuring the maximum fluorescence intensity in these ROIs. Data were normalized to the mean of the DMSO control of each group.

**H-89 assay**. HeLa cells were plated on 12 mm coverslips and incubated the following day for 30 min with 100 μM H-89 (Cayman Chemicals, 10010556) in DMSO or DMSO alone (vehicle) and H-89 was washed out with fresh medium for 5 min. Cells were fixed with 4% paraformaldehyde for 15 min at RT and permeabilized with 100% methanol at −20 °C for 15 min prior to immunostaining with ERGIC53 mouse monoclonal antibody (G1/93 or OTI1A8) before epifluorescence imaging, as described for the BFA assay. Cells were analyzed using Fiji and the number of ERGIC53-positive spots was quantified with the Spot Counter plugin. Data were normalized to the mean fluorescence of the DMSO control of each group. Data were analyzed with a Mann-Whitney U non-parametric test.

**Transmission electron microscopy**. Fibroblasts were grown in 12-well plates and fixed with 2% glutaraldehyde (Sigma-Aldrich, G5882) in PB (0.1 M phosphate buffer, pH 7.4) for 60 min at RT. Subsequently, cells were washed four times with PB and post-fixed with 1% osmium tetroxide and 1% potassium ferrocyanide in PB for 60 min at RT. Then, cells were again washed four times with PB and four times with water. Cells were incubated overnight in 0.5% uranyl acetate and dehydrated with graded steps of ethanol (30%, 50%, 70%, 96%, 100%) and embedded in Epon resin. Then, 70 nm sections were stained with 2% uranyl acetate solution and lead citrate solution. Stained sections were then examined using a CM12 transmission electron microscope (Phillips).

**Live-cell epifluorescence microscopy**. Cells were seeded in four-compartment dishes (Greiner 627870) and transfected as described above (3:1 weight ratio, reporter construct:Golgi label). Before imaging, the culture medium was exchanged for Leibovitz's L-15 (Gibco 21083027). Samples were imaged using a DMI6000B (Leica Microsystems) with a heated stage (Pecon) and objective heater. All samples were imaged using an HC PL APO 63x/1.40–0.60 OIL objective. VSVG-ts045-EGFP experiments were performed at 32 °C after overnight incubation at 40 °C, while all other epifluorescence experiments were performed at 37 °C. For RUSH experiments, an equal amount of Leibovitz's L-15 supplemented with biotin was added to the well immediately before imaging, to reach a final concentration of 40 μM biotin. Live-cell imaging was started immediately with 15 or 30 s frame rates. Analysis was performed with Fiji, after registration of the image stacks, the increase in fluorescence was measured in the Golgi area by using the thresholded mScarlet-Giantin signal as an image mask.

**FRET-FLIM**. All imaging took place in Leibovitz's L-15 supplemented with 10 μg/mL cycloheximide (Sigma-Aldrich, C4859) and cells were pulsed with biotin as described above. Imaging was performed on a Leica SP8 SMD system at 37 °C, equipped with an HC PL APO CS2 63x/1.20 Water objective. Fluorophores were excited with a pulsed white-light laser, operating at 80 MHz. mCitrine was excited at 514 nm, two separate HyD detectors were used to collect photons, set at 521–565 nm and 613–668 nm, respectively. Photons were collected for 1 min and lifetime histograms of the donor fluorophore were fitted with monoexponential decay functions convoluted with the microscope instrument response function in Leica LAS X.

**SDS-PAGE and immunoblotting**. Cells were plated in 12-well plates in culture medium and lysed the following day with SDS lysis buffer (1% SDS, 10 mM Tris-HCl pH 6.8). Lysates were diluted to equal protein content (30 μg per lane) with SDS lysis buffer, separated with SDS-PAGE on 4–20% Mini-PROTEAN TGX

Precast Protein Gels (Biorad, 4561094) and subsequently transferred onto 0.45 μm PVDF membranes. Small-molecular weight proteins (Bet1 and Bet1L) were separated on 16% Schaegger gels[88].

**Quantification and statistical analysis**. All mean values represent the average of all cells analyzed. All comparisons between two groups were first checked for similar mean and median values and acceptable (<3x) difference in variance, before statistical analysis with an unpaired two-sided Student's t-test. Relative intensity data were first transformed using the binary logarithm before analysis with an unpaired two-sided Student's t-test. H-89 data were analyzed with a Mann-Whitney U non-parametric test. BFA washout data were analyzed with an unpaired two-sided Student's t-test between healthy donors and Stx5M55V patients followed by Bonferroni correction for multiple comparisons. Stx5 kinetics data were analyzed with a one-way ANOVA, followed by a post hoc Tukey's honestly significant difference test. FRET-FLIM data were averaged per experiment before analysis with a paired Student's t-test. $P < 0.05$ was considered significant. $^*P < 0.05$, $^{**}P < 0.01$, $^{***}P < 0.001$, $^{****}P \leq 0.0001$. All statistical analyses were performed using R statistical software. All numerical data were visualized using R package ggplot2[89], with violins representing the overall distribution of the data and mean ± 95% CI overlaid.

**Reporting summary**. Further information on research design is available in the Nature Research Reporting Summary linked to this article.

## Data availability
The primary microscopy data generated in this study have been deposited in the Zenodo database under accession code 10.5281/zenodo.5440334 (https://doi.org/10.5281/zenodo.5440334). The microarray and exome sequencing data are available under restricted access for patient privacy reasons; access can be obtained upon request. The processed microscopy data and complete western blots generated in this study are provided in the Source Data file. Source data are provided with this paper.

## Code availability
ImageJ macros for quantification of the RUSH experiments are available upon request to the corresponding authors.

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

## Acknowledgements
We thank the family for participating in this study. We thank the following people for constructs: Hesso Farhan and Franck Perez (Str-KDEL_ManII-SBP-EGFP; Addgene plasmid #65252), Jennifer Lippincott-Schwartz (pEGFP-VSVG; Addgene plasmid #11912), Feng Zhang (pSpCas9n(BB)-2A-Puro (PX462) V2.0; Addgene plasmid #62987) and Dorus Gadella (pmScarlet-i_Giantin_C1, Addgene plasmid #85050). We also thank the Microscopic Imaging Center of the Radboud Institute for Molecular Life Sciences for use of their microscopy facilities. N.H.R. is funded by a Long-Term Fellowship from the European Molecular Biology Organization (EMBO-LTF, ALTF 232-2016) and a Veni grant from the Netherlands Organization for Scientific Research (016.VENI.171.097). G.v.d.B. is funded by a Young Investigator Grant from the Human Frontier Science Program (HFSP; RGY0080/2018) and a Vidi grant from the Netherlands Organisation for Scientific Research (NWO-ALW VIDI 864.14.001). G.v.d.B has also received funding from the European Research Council (ERC) under the European Union's Horizon 2020 research and innovation program (grant agreement No. 862137). D.J.L. is funded by a Vidi grant (ZONMW VIDI 917.13.359), a ZONMW Medium Investment Grant (40-00506-98-9001) from the Netherlands Organisation for Scientific Research, and E-rare grants EUROCDG2 and EUROGLYCAN-omics. EUROGLYCAN-omics was supported by the Netherlands Organisation for Health Research and Development, grant nr 90030376501, under the frame of E-rare-3, the ERA-Net for research on rare diseases. K.Õ., M.-A.V., S.P. and K.M. were supported by the Estonian Research Council grants GARLA8175, PUT355, PUTJD827, and PRG471.

## Author contributions
P.T.A.L., M.t.B., D.J.L. and G.v.d.B. designed the experiments and wrote the paper. E.C.F.G. contributed to the Stx5 kinetics, and FLIM experiments. A.A., M.-A.V., M.B., F.Z., K.H., K.R., K.M. and K.Õ. contributed to the clinical data, exome sequencing, and glycomics. O.F. and S.P. performed homozygosity mapping and prioritization of exome variants. N.H.R. and R.d.B. performed TEM. R.A. contributed to the Stx5ΔL experiments. P.T.A.L. and M.t.B. performed all other experiments. All authors contributed to writing the paper.

## Competing interests
The authors declare no competing interests.
