## [Peer Review File. · Nature Communications]

Congenital disorder of glycosylation caused by starting site-specific variant in syntaxin-5Reviewers' Comments:

Reviewer #1:

Remarks to the Author:

In this manuscript, the authors aim to describe the mechanistic underpinnings of a mutation identified in patients with severe, multi-system, fatal disease. The underlying mutation is in the SNARE protein syntaxin-5 (Stx5), which functions in membrane fusion events in ER-Golgi trafficking as well as in retrograde trafficking between Golgi stacks. Patients completely lack the short isoform of Stx5, as the mutation impacts an alternative start codon at amino acid 55, resulting in pleiotropic defects and ultimately fatal liver failure before the end of the first year of life. While the characterization of the disease-causing mutation ultimately warrants publication and patient cells present a unique opportunity to investigate isoform specific functions, several issues need to be resolved before the manuscript is ready for publication.

Chiefly, a main argument/idea set forth in the paper is that the long and short Stx5 isoforms have distinct functions and that the long isoform alone cannot perform all the functions of the short isoform, leading to the clinical outcomes observed in the patients. An alternative explanation is that there is simply insufficient total Stx5 when one isoform is lost, which underlies the patient phenotypes. Primary dermal fibroblasts from the patients show that Stx5 Short is absent and Stx5 Long levels are unchanged. Based on the western blot in Figure 2B, it appears that Stx5 Short comprises ~50% of the total Stx5 in control cells. It seems plausible that loss of ~50% of Stx5- an essential gene with a critical function across all cell types- would have broad physiological consequences and potentially be incompatible with life. This ratio may also differ across cell types and lead to a further decrease in total Stx5 relative to control cells of the same cell type. For example, if liver cells have an even higher proportion of Stx5S and consequently lose an even higher percentage of their total Stx5 as a result of the mutation, one could envision this leading to the severe liver phenotypes identified in the patients.

Furthermore, it is not always clear what isoform-specific role is being proposed. For example, in line 284 in the discussion, it is stated that Stx5 S is more important for anterograde trafficking, then just a few lines later in line 288, the dominant role of Stx5 is in intra-Golgi trafficking (presumably retrograde). These inconsistent suggestions of function are present throughout the manuscript. Additionally, 'retrograde'/'anterograde' trafficking are used frequently without always being clear if retrograde means Golgi-ER transport or retrograde transport within Golgi stacks.

Regarding isoform-specific roles:

1. To determine if there truly are isoform-specific roles for Stx5 Long and Short, one could increase levels of Stx5 Long in patient cells lacking Stx5 short. While these experiments may be informative, they may also be problematic in that then Stx5 Long may perform functions of Stx5 Short even if it does not normally do so. Perhaps an alternative approach would be to reduce total Stx5 levels by ~50% and show that the defect is different/absent as compared to loss of Stx5 short only. This could be done with a heterozygous cell line lacking one copy of Stx5. It would be critical to perform these comparison experiments in the same cell type across all conditions and confirm levels across Stx5 cell types by western blot. As the defects shown by the lectin assay were quite striking (2d,g), this might be a good assay to examine rescue with different amounts/isoforms of Stx5. If cells are available from the maternal carrier, it may also be interesting to examine these as well to as they would be expected to have half the level of Stx5 Short and a normal amount of Stx5 Long.

2. There does seem to be a slight difference in localization of Stx5 L and S, with Stx5 L more efficiently returned to the ER, based on the data presented in Figures 6 and 7. This difference in localization, although suggested previously, was not very well documented and worth presenting here. As data in Figures 6 and 7 comes only from transient transfections of tagged proteins and there is no verification that these constructs are expressed to the same level or are functional, it is important to have additional verification of the somewhat distinct distribution of the long and short isoforms. This subtle difference in isoform localization is also suggested by a change in distribution of Stx5 in the patient cells that lack the short isoform, however, these data are dispersed across several figures,

thus difficult to follow, and one could also envision that lack of the short isoform impacts the localization of the long isoform-especially if it is truly involved in retrograde Golgi-ER transport as suggested in line 195.

Ideally, one could CRISPR-edit the endogenous Stx5 locus to generate 1 specific isoform that is tagged with one color and introduce the other isoform into another locus. One would then need to verify that both isoforms are expressed at levels comparable to unedited cells and that the tags do not impact the function of Stx5.

On a related note, in the experiments where Stx5 isoforms are tagged at the c-terminus, how do the authors know there is not alternative translation from the long construct, generating some of the short isoform? Does the long construct have the patient mutation to ensure that only the long isoform is generated?

3. It does not seem appropriate to make direct comparisons between HeLa cells CRISPR edited to lose only the long isoform and the patient cells which lose only the short isoform (Figure 5). Ideally one could CRISPR edit the patient derived cells to both correct the original mutation and generate a mutation resulting in loss of the long isoform. If this is not possible, generate each isoform specific cell line in the same background and ensure that the phenotypes observed in patient cells are confirmed. HeLa cells seem challenging and not ideal as they already contain too many chromosomes and the levels of total Stx5 may already be abnormal. There is no data regarding the baseline levels of the 2 Stx5 isoforms in HeLa cells or the CRISPR-edited cells. Additionally, it is unclear what this BFA experiment is intended to show. The text argues that it shows a role for Stx5S in retrograde Golgi-ER transport, but the figure is also consistent with a role for Stx5S in ER-Golgi transport.

A second major issue is that much emphasis is placed on altered glycosylation in plasma from patients, diminished cell surface glycosylation in patient cells, mis-localized glycosyltransferases in patient cells, and the relationship between these defects and the clinical presentation of the patients. While there is clearly a difference in the pattern of glycosylation in the patient plasma samples and at the cell surface (Figures 1g,h, 2d-g), and these defects likely have clinical consequences, it seems like a more general defect in protein transport could underlie the glycosylation defect and be a more direct cause of disease.

In regard to the analysis of glycosyltransferase localization (Figure 3), it is difficult to follow. It would be helpful to have a diagram depicting where each enzyme is expected to localize in the Golgi and the steps in the glycosylation pathway. Additionally, it would have been helpful to see GM130 and TGN46 together in each image, or at least in panels next to each. Perhaps a table summarizing the changes in the localization would be helpful to identify patterns. Additionally, if there is less TGN46 overall (Figure 4h), is diminished colocalization with TGN46 meaningful? Also, while strong statistical significance is shown for alterations in many of the glycosyltransferases, likely due to a very large sample size, I'm not sure that any of the changes are biologically meaningful when the most extreme change in the PCC value between control and patient samples is on the order of 0.1. It's also not clear how this data was analyzed as no description is provided in the methods, so it is difficult to discern whether it was done appropriately. Finally, are the overall levels of glycosyltransferases impacted and further complicating analysis?

Minor comments/suggestions

- the last sentences of the introduction and the beginning of the discussion highlight the novelty of disease caused by a mutation in an alternative start site. While this may be the first described instance where loss of a short isoform due to change in an alternative start site causes disease, there are many previous descriptions of alternative start site variations leading to disease (see Bogaert et al, N-Terminal Proteoforms in Human Disease, 2020 for review)
- the images in the main figures are too small to evaluate the interpretation in the text for Figures 3-7. Insets are needed.
- the description of SNARE complex formation in the first paragraph of the introduction is a bit hard to follow. Perhaps this can be clarified or schematized in a model.

-the statement in the introduction that yeast Sed5p corresponds to Stx5 Long contradicts the authors' 2019 review. Please verify that the text is correct.

-more description and applicable references for CDG screening and how the results here compare to COG disorders would be helpful

-To substantiate the claim that TGN46 levels are reduced (Figure 4h), this result should be quantified. If TGN46 levels are reduced, would you expect a more dramatic defect in Golgi architecture (Figure S5)?

-if there is a change in overall shape/staining of any component (GM130, TGN46, Stx5 overall) is comparing the Pearson Correlation Coefficient across cell types the appropriate way to analyze?

-In 4C, why would there be a difference in overall b-COP staining if levels are unchanged (4g)?

-7d hard to believe this is really a significant difference

-For Figure 6, the legend title and title of 6c are confusing. I think this is supposed to highlight that Stx5L returns to the Golgi, so maybe 'Faster Golgi exit'? Also, if the rate of Golgi exit is faster for Stx5L, wouldn't we expect this value to be higher in 6d? It's not clear how this was quantified.

Reviewer #2:

Remarks to the Author:

This manuscript by Linters et al identifies a new mechanism of disease in humans, due to a pathogenic mutation in Stx5, an ER-Golgi SNARE protein. Affected individuals have a fatal multisystemic disease, liver failure and skeletal abnormalities, and abnormal glycosylation. The pathogenic variant in Stx5 is in an alternative starting methionine of syntaxin-5, resulting in complete loss of the shorter protein isoform of syntaxin-5 (Stx5S), whilst the longer protein isoform (Stx5L) remains unchanged.

Linters et al undertake a detailed molecular investigation in patient fibroblasts and gene-edited HeLa cells, to determine the functional consequences of loss of Stx5S. A range of well-established molecular tools were utilised to study ER-Golgi trafficking (eg. RUSH assay, tsVSV-GFP trafficking) and determine transient protein interactions (FRET and FRET-FLIM). The authors demonstrated clear impairments in Golgi trafficking in these cell lines in some assays, and modest impairments in other endpoints. However, when all the results are considered collectively, it provides compelling evidence that the variant is likely pathogenic.

This manuscript is of interest to clinicians and geneticists, since it expands the clinical spectrum of genetic congenital disorders of glycosylation, and it suggests that mutations in Stx5 should be considered when screening patients with glycosylation abnormalities. Additionally, it is the first report for a mutation in an alternative starting methionine, causing protein isoform-specific effects.

I am happy that the authors have provided sufficient details in the methods sections to allow reproduction of their research. The statistical analysis is appropriate. The manuscript is well written, with appropriate background information, and a clear and concise discussion. The figures are clear, and well presented.

This manuscript would also have interest to the more general scientific field. This is because the authors also provide new evidence on the molecular function of Stx5S. Previously, it was thought that Stx5L was more critical to early Golgi trafficking, whilst Stx5S was more involved in late Golgi trafficking. Instead, the authors provide evidence that Stx5S is more important for anterograde trafficking, and loss of Stx5S in patient fibroblasts alters distribution of trafficking proteins and glycosylation enzymes. Additionally, loss of Stx5L indicated that Stx5L is more involved in retrograde trafficking. The authors also provide some additional information on the location-specific interaction of Stx5L and Stx5S with Bet1L, a known Golgi intra-trafficking protein. These additional insights would be of interest to researchers outside the human genetics field, as it provides fundamental insights into Stx5 protein function.

Paternal DNA was unavailable for genetic testing, and an unaffected sibling was also not tested. Given that Stx5 is a new gene and the affected individuals are homozygous for the variant, it would be good to know if the parents were consanguineous. If the parents are not consanguineous or not known to be, there would need to be some discussion about potential reasons for the variant being homozygous. Are the parents from the same small town or small religious group? Could it be the variant in the probands is hemizygous (ie one allele with the variant inherited from the mother and the gene partially or totally deleted in the paternally inherited allele)? It would be useful to include additional information about the ethnicity of the family.

Reviewer #3:

Remarks to the Author:

NCOMMS-20-14459-T

Congenital disorder of glycosylation caused by starting site specific variant in syntaxin-5

Peter Linders, Eveline Gerretsen, Angel Ashikov, Mari-Anne Vals, Natalia Revelo, Richard Arts, Melissa Baerenfaenger, Fokje Zijlstra, Karin Huijben, Kimiyo Raymond, Kai Muru, Olga Fjodorova, Sander Pajusalu, Katrin Ounap, Martin ter Beest, Dirk Lefeber, and Geert Bogaart

In this study, Linders et al. discovered a mutation of the Golgi SNARE Syntaxin 5 as the cause of a novel and severe Congenital Disorder of Glycosylation (CDG), which is characterized by high abortion rate and badly skeletal dysplasia and abnormal facial features. Syntaxin 5 exists as two forms, long and short forms, by alternative translational initiation from the same mRNA. Interestingly, the mutation occurs at the second initiation codon, which blocks the production of the short form of Syntaxin 5. The authors showed that this unusual mutation induces mislocalization of several glycosyltransferases in the Golgi apparatus and thus leads to severe defects in protein glycosylation. They also perform cell biology assays to show trafficking defects in the mutant cells. Overall, the study is of high interest and physiologically relevant, but the quality of the data is poor, and the proposed molecular mechanism is not convincing. Specific points:

1. Figure 1d,e, can the different samples be analyzed in parallel and so one can directly compare the amount of each individual form? Figure 1g,h, since the y-axis is shown in a different scale, is there a difference in the amount of transferrin in control and patient serum?
2. Supplementary Table 3, what is the explanation of the differences between IV:9 and IV:9-10? Such a concern exists in multiple figures, which make the conclusions questionable.
3. Figure 2b,c, STX5 has been shown as an essential gene in yeast, are the patient fibroblasts viable? Are these technical replicates or different cell lines? The level of some proteins in the same cell type varies too much. What's the explanation of decreased Bet1L level in the patient fibroblasts? Figure 2d, the authors may want to increase DAPI intensity to make it more visible. Given that the lectin signal is normally not linear, a FACS analysis of the same cells would be more reliable.
4. Figure 3 and Supplementary Figure 4, the conclusion that "Glycosylation enzymes mislocalize in Stx5M55V patient fibroblasts" is invalid. Figure 3a, the morphology of the Golgi and the relative levels of the two colors are very different in the two cell lines, this will inevitably affect the quantitation results. Similar concerns exist in all following panels. The authors showed that different Golgi enzymes behave very differently, although they are all single-pass type II membrane proteins. What is the explanations? Does Stx5S binds some but not other enzymes? Figure 3c, the specificity of MAN2A1 is very poor, so the conclusion is not convincing. All proteins used in the study should be tested by Western blots as in Figure 2b.
5. Supplementary Figure 5a, the conclusion that Stx5M55V does not affect the Golgi structure is not

convincing. The morphology of the Golgi is very different between the two cell lines, different Golgi morphologies are also seen in Figure 3 and Supplementary Figure 4 even in different panels of the mutant cells. The EM image also indicated part of swollen Golgi cisternae is in a Stx5M55V patient cell. A gallery and quantification of the EM images should be included.

6. Figure 4, the quality of the images is of low quality due to the high background. The quality of the Western blots in Figure 4g,h is also very poor, with large variations within the same cell types (e.g. the two control lanes), raising a concern of the quality of the work.

7. Figure 5, the functional assay is incomplete and not convincing. In addition to inhibiting ARF and COPI vesicle formation, BFA also modifies membrane lipids, and so the results does not provide much useful information. Although BFA-washout can be used to test anterograde trafficking. In this experiment, live cell imaging would be more convincing. For fixed cell imaging, multiple time points are needed. Additional Golgi and ER markers (e.g., PDI in Fig S6) should be included.

8. Figure 6 and 7, because of the inconsistent fluorescence intensity and the poor quality of images, the conclusion is invalid. This leaves no mechanism in this study.

9. The author should perform rescue experiments in the cell based experiments by expressing Stx5S.

10. In many figures, different scales or scale bars are used in control vs. mutant cells. This is not acceptable.

11. Some basic information is missing. For example, what antibodies are used for Syntaxin 5? Do the authors have an antibody specific for Stx5L?

12. There are typos and grammar mistakes in the text. A few example: page 9/37 line 221, "...was faster for Stx5L-SBP-mCitrine than for than for Stx5S or Stx5L Δ ER" should be "...was faster for Stx5L-SBP-mCitrine than that for Stx5S or Stx5L Δ ER". On page 9/37 line 230, "either without or with 30 mins biotin" should be either without or with biotin for 30 mins.

We appreciate the feedback of all reviewers that helped us to improve our manuscript. We have extensively revised our manuscript, and it includes the following new data:

- We generated a fibroblast cell line lacking Stx5L with CRISPR/Cas9 and show that the effect on glycosylation is much milder than with the loss of Stx5S.
- We show with extensive transmission electron microscopy that the morphology of the Golgi is strongly affected in the Stx5M55V patient fibroblasts.
- We have added several new ER-Golgi trafficking assays in the patient fibroblasts, including Brefeldin A washout assays and experiments with temperature-sensitive VSVG-EGFP synchronized trafficking, both strengthening our conclusion that intra-Golgi trafficking is strongly affected in Stx5M55V patient cells.
- We have performed new immunostainings with a Stx5L-specific antibody, confirming that Stx5L localizes more to the *cis*-Golgi in Stx5M55V patient fibroblasts

Together, we believe we have answered the concerns of the reviewers adequately and hope that the manuscript meets the criteria for publication. Below, we elaborate on each concern from the reviewers.

Reviewer #1 (Remarks to the Author):

In this manuscript, the authors aim to describe the mechanistic underpinnings of a mutation identified in patients with severe, multi-system, fatal disease. The underlying mutation is in the SNARE protein syntaxin-5 (Stx5), which functions in membrane fusion events in ER-Golgi trafficking as well as in retrograde trafficking between Golgi stacks. Patients completely lack the short isoform of Stx5, as the mutation impacts an alternative start codon at amino acid 55, resulting in pleiotropic defects and ultimately fatal liver failure before the end of the first year of life. While the characterization of the disease-causing mutation ultimately warrants publication and patient cells present a unique opportunity to investigate isoform specific functions, several issues need to be resolved before the manuscript is ready for publication.

#1.1 Chiefly, a main argument/idea set forth in the paper is that the long and short Stx5 isoforms have distinct functions and that the long isoform alone cannot perform all the functions of the short isoform, leading to the clinical outcomes observed in the patients. An alternative explanation is that there is simply insufficient total Stx5 when one isoform is lost, which underlies the patient phenotypes. Primary dermal fibroblasts from the patients show that Stx5 Short is absent and Stx5 Long levels are unchanged. Based on the western blot in Figure 2B, it appears that Stx5 Short comprises ~50% of the total Stx5 in control cells. It seems plausible that loss of ~50% of Stx5- an essential gene with a critical function across all cell types- would have broad physiological consequences and potentially be incompatible with life. This ratio may also differ across cell types and lead to a further decrease in total Stx5 relative to control cells of the same cell type. For example, if liver cells have an even higher proportion of Stx5S and consequently lose an even higher percentage of their total Stx5 as a result of the mutation, one could envision this leading to the severe liver phenotypes identified in the patients.

We thank the reviewer for this concern. To dissect whether the observed effects of the STX5 mutation in the patient fibroblasts is a result of the specific loss of the Stx5S isoform, or

rather the result of lower protein levels of total Stx5, we performed several new experiments. Two lines of evidence show that the ratio of the Stx5S and L isoforms might be more important for Golgi transport than their individual expression levels:

1. We now show expression of Stx5S and Stx5L in primary mononuclear cells from peripheral blood from four healthy individuals. The data show that while overall Stx5 expression levels vary more than 2-fold between individuals, the ratio between Stx5S and Stx5L is highly similar among individuals, and approximately equimolar for all individuals (new **Supplementary Figure 4**).
2. We generated fibroblasts with CRISPR/Cas9 knockout of Stx5L (new **Supplementary Figure 5**). By our knowledge, this is the first example of an isoform-specific knockout in primary dermal fibroblasts. However, as it is technically difficult to obtain perfect single clones from fibroblast cultures, we still observed a small residual band of Stx5L by Western blot (new **Supplementary Figure 5a**). In contrast to the strongly reduced and non-uniform SNA-I labeling of Stx5M55V patient fibroblasts (that only have Stx5L but not Stx5S), Stx5 Δ L fibroblasts still showed pronounced plasma membrane labeling with SNA-I with a staining pattern comparable to healthy donor fibroblasts. In addition, PNA binding to Stx5M55V patient fibroblasts was strongly increased, but was not consistently altered in Stx5 Δ L fibroblasts. These findings show that both O- and N-glycosylation are more affected upon loss of Stx5S than upon loss of Stx5L.

It might very well be that the ratio of Stx5L over Stx5S differs between cell types, as suggested by the reviewer. Indeed, as we now explain in the Discussion section of our revised manuscript (**page 13**), Western blot revealed different ratios of the two Stx5 isoforms in different organs in mice (ref 1), suggesting that (i) different cell types express different levels of Stx5S and Stx5L and this is probably related to their exocytic function, and (ii) the initiation of starting translation might be regulated and not merely dependent on the binding affinities of the ribosome. This implies that the loss of Stx5S might very well affect different cell types differently, and because of this it is challenging to directly extrapolate our results from patient fibroblasts to the clinical phenotype. Nevertheless, our data show that indeed the ratio of the isoforms is the most important determinant of Stx5 function, rather than total Stx5 levels.

- (1) Hui, N, N Nakamura, B Sönnichsen, D T Shima, T Nilsson, and G Warren. "An Isoform of the Golgi T-SNARE, Syntaxin 5, with an Endoplasmic Reticulum Retrieval Signal." *Molecular Biology of the Cell* 8, no. 9 (September 1997): 1777–87.

#1.2 - Furthermore, it is not always clear what isoform-specific role is being proposed. For example, in line 284 in the discussion, it is stated that Stx5 S is more important for anterograde trafficking, then just a few lines later in line 288, the dominant role of Stx5 is in intra-Golgi trafficking (presumably retrograde). These inconsistent suggestions of function are present throughout the manuscript. Additionally, 'retrograde'/'anterograde' trafficking are used frequently without always being clear if retrograde means Golgi-ER transport or retrograde transport within Golgi stacks.

We agree with the reviewer that this wording is inconsistent. We have extensively revised our manuscript and removed these inconsistencies. We now consistently explain throughout the revised manuscript that the phenotype of the Stx5M55V cells is attributable to the role of Stx5S in retrograde Golgi-ER and intra-Golgi trafficking. A list of all changed inconsistencies including their line numbers in the revised manuscript:

- Line 77: “the loss of Stx5S leads to defects in anterograde Golgi trafficking” changed to “the loss of Stx5S leads to defects in intra-Golgi trafficking”
- Line 80: “identify Stx5S as the dominant Qa-SNARE for intra-Golgi transport” changed to “identify Stx5S as the dominant Qa-SNARE for retrograde Golgi-ER and intra-Golgi transport”
- Line 219, caption: “Stx5 mediates retrograde Golgi trafficking” changed to “Stx5 mediates retrograde Golgi-ER and intra-Golgi trafficking”
- Line 264: “To further delineate the role of Stx5L in anterograde ER-Golgi trafficking” changed to “To further delineate the role of Stx5L in retrograde Golgi-ER trafficking”
- Line 267: “Further investigation of anterograde trafficking” changed to “Further investigation of anterograde ER-Golgi trafficking”
- Line 351: “both Stx5 isoforms can mediate both early and late anterograde and retrograde trafficking” changed to “both Stx5 isoforms can mediate both early and late anterograde and retrograde Golgi trafficking”
- Line 353: “the role of Stx5S is more important for anterograde trafficking” changed to “the role of Stx5S is more important for retrograde Golgi-ER and intra-Golgi trafficking”
- Line 360: “the dominant role of Stx5S in intra-Golgi trafficking” changed to “the dominant role of Stx5S in retrograde Golgi-ER and intra-Golgi trafficking”

Regarding isoform-specific roles:

#1.3 - To determine if there truly are isoform-specific roles for Stx5 Long and Short, one could increase levels of Stx5 Long in patient cells lacking Stx5 short. While these experiments may be informative, they may also be problematic in that then Stx5 Long may perform functions of Stx5 Short even if it does not normally do so. Perhaps an alternative approach would be to reduce total Stx5 levels by ~50% and show that the defect is different/absent as compared to loss of Stx5 short only. This could be done with a heterozygous cell line lacking one copy of Stx5. It would be critical to perform these comparison experiments in the same cell type across all conditions and confirm levels across Stx5 cell types by western blot. As the defects shown by the lectin assay were quite striking (2d,g), this might be a good assay to examine rescue with different amounts/isoforms of Stx5. If cells are available from the maternal carrier, it may also be interesting to examine these as well to as they would be expected to have half the level of Stx5 Short and a normal amount of Stx5 Long.

We thank the reviewer for this analysis and constructive feedback. Unfortunately, no material from the maternal carrier was available. The results we show at point #1.1 demonstrate that modulating the ratio of Stx5S and Stx5L gives different results with regards to glycosylation, and functional readouts with BFA washout (Figure 5, Supplementary Figure 14), VSVG (Figure 5, Supplementary Figure 14), RUSH (Figure 6, Supplementary Figure 15) and FLIM (Figure 7, Supplementary Figure 17) show different roles for either isoform in Golgi trafficking.

In fact, we attribute our inability to conduct the rescue experiments requested by reviewer #3 to this (see point #3.13): for successful rescue, similar levels of expression of Stx5L and S are likely needed which are difficult to attain.

#1.4 - There does seem to be a slight difference in localization of Stx5 L and S, with Stx5 L more efficiently returned to the ER, based on the data presented in Figures 6 and 7. This difference in localization, although suggested previously, was not very well documented and worth presenting here. As data in Figures 6 and 7 comes only from transient transfections of tagged proteins and there is no verification that these constructs are expressed to the same level or are functional, it is important to have additional verification of the somewhat distinct distribution of the long and short isoforms. This subtle difference in isoform localization is also suggested by a change in distribution of Stx5 in the patient cells that lack the short isoform, however, these data are dispersed across several figures, thus difficult to follow, and one could also envision that lack of the short isoform impacts the localization of the long isoform-especially if it is truly involved in retrograde Golgi-ER transport as suggested in line 195.

We performed new experiments with an antibody specific to Stx5L (antibody validation for immunofluorescence in new Supplementary Figure 13) in immunofluorescence microscopy experiments to compare the distribution of Stx5L in both patient and healthy cells. The results, presented in new Figure 4d-g, confirm that Stx5L localizes more to the *cis*-Golgi and ER in patient cells.

#1.5 - Ideally, one could CRISPR-edit the endogenous Stx5 locus to generate 1 specific isoform that is tagged with one color and introduce the other isoform into another locus. One would then need to verify that both isoforms are expressed at levels comparable to unedited cells and that the tags do not impact the function of Stx5.

We appreciate the idea from the reviewer, but unfortunately, we did not manage to genomically tag endogenous Stx5 with CRISPR/Cas9. However, we distinguished the subcellular localization of Stx5L using an isoform-specific antibody as explained at point #1.4.

#1.6 - On a related note, in the experiments where Stx5 isoforms are tagged at the c-terminus, how do the authors know there is not alternative translation from the long construct, generating some of the short isoform? Does the long construct have the patient mutation to ensure that only the long isoform is generated?

For all experiments with heterologous expression of Stx5L, we used the long construct carrying the patient mutation. Therefore, the short isoform cannot be produced when this construct is expressed. We now emphasize this at pages 10 and 17 of our revised manuscript. Additionally, the constructs were deposited to Addgene and will be available upon publication.

#1.7 - It does not seem appropriate to make direct comparisons between HeLa cells CRISPR edited to lose only the long isoform and the patient cells which lose only the short isoform (Figure 5). Ideally one could CRISPR edit the patient derived cells to both correct the original

mutation and generate a mutation resulting in loss of the long isoform. If this is not possible, generate each isoform specific cell line in the same background and ensure that the phenotypes observed in patient cells are confirmed. HeLa cells seem challenging and not ideal as they already contain too many chromosomes and the levels of total Stx5 may already be abnormal. There is no data regarding the baseline levels of the 2 Stx5 isoforms in HeLa cells or the CRISPR-edited cells.

As requested by the reviewer, we generated Stx5 Δ L fibroblasts as explained at point #1.3, to be able to perform the lectin stainings in cells with a similar background (Stx5M55V patient fibroblasts and Stx5 Δ L fibroblasts). As requested, we now also include Western blot results from the Stx5 Δ L HeLa cells (new **Supplementary Figure 5f**).

#1.8 - Additionally, it is unclear what this BFA experiment is intended to show. The text argues that it shows a role for Stx5S in retrograde Golgi-ER transport, but the figure is also consistent with a role for Stx5S in ER-Golgi transport.

We have extended the BFA experiment by performing a washout of the drug. We now show that the loss of Stx5S causes an incomplete redistribution of GALNT2 from the Golgi to the ER (during the BFA incubations), but the re-localization of GALNT2 back to the Golgi (upon BFA washout) occurs more rapidly than in control cells. This is consistent with the idea that BFA affects Stx5S-mediated retrograde Golgi-ER transport, while Stx5L can still perform ER-Golgi anterograde trafficking in the Stx5M55V patient fibroblasts. Moreover, we performed VSVG trafficking experiments in the patient fibroblasts. The data show that, although VSVG is trafficked normally from the ER to the Golgi in patient cells, plasma membrane localization of VSVG was significantly decreased compared to healthy cells. These new data are in line with our model of a role of Stx5S in intra-Golgi trafficking and are presented at **page 10, figure 5e-g, Supplementary Figure 14d** of our revised manuscript.

#1.9 - A second major issue is that much emphasis is placed on altered glycosylation in plasma from patients, diminished cell surface glycosylation in patient cells, mis-localized glycosyltransferases in patient cells, and the relationship between these defects and the clinical presentation of the patients. While there is clearly a difference in the pattern of glycosylation in the patient plasma samples and at the cell surface (Figures 1g,h, 2d-g), and these defects likely have clinical consequences, it seems like a more general defect in protein transport could underlie the glycosylation defect and be a more direct cause of disease.

We fully agree with the reviewer that glycosylation abnormalities likely are not causing the clinical symptoms to a large extent and apologize if this was not clear in the previous version of our manuscript. Nevertheless, glycosylation abnormalities were the starting point to identify the mutation in *STX5* and we used glycosylation read-outs in fibroblasts to confirm the suitability of fibroblasts as a model system. Indeed, we then continue with cell biological readouts to identify additional, possibly causative, biological abnormalities. We now more elaborately discuss in our Discussion section that the wide variety of symptoms in the Stx5M55V patients is likely not only caused by aberrant glycosylation but by a general defect in exocytic trafficking (**page 14**).

#1.10 - In regard to the analysis of glycosyltransferase localization (Figure 3), it is difficult to follow. It would be helpful to have a diagram depicting where each enzyme is expected to localize in the Golgi and the steps in the glycosylation pathway. Additionally, it would have been helpful to see GM130 and TGN46 together in each image, or at least in panels next to each. Perhaps a table summarizing the changes in the localization would be helpful to identify patterns. Additionally, if there is less TGN46 overall (Figure 4h), is diminished colocalization with TGN46 meaningful?

We thank the reviewer for this suggestion and now include diagrams depicting the physiological localization of each glycosyltransferase within the Golgi for all panels in **Figure 3** and **Supplementary Figures 6 and 7**. Moreover, we rearranged the panels in such a way that the co-staining with GM130 and TGN46 are displayed side-by-side (**Fig. 3a-d**). Lastly, we addressed the comment on using TGN46 as a *trans*-Golgi marker by performing colocalization experiments with another *trans*-Golgi marker, p230 (ref. 1). We observed similar patterns of colocalization between the tested glycosyltransferases and TGN46 or p230 (new **Supplementary Figure 7**).

- (1) Gleeson, P. A., T. J. Anderson, J. L. Stow, G. Griffiths, B. H. Toh, and F. Matheson. "P230 Is Associated with Vesicles Budding from the Trans-Golgi Network." *Journal of Cell Science* 109, no. 12 (December 1, 1996): 2811–21.

#1.11 - Also, while strong statistical significance is shown for alterations in many of the glycosyltransferases, likely due to a very large sample size, I'm not sure that any of the changes are biologically meaningful when the most extreme change in the PCC value between control and patient samples is on the order of 0.1. It's also not clear how this data was analyzed as no description is provided in the methods, so it is difficult to discern whether it was done appropriately. Finally, are the overall levels of glycosyltransferases impacted and further complicating analysis?

We now include a description of how the colocalization analysis was performed in the methods section, under the subsection Immunofluorescence (**page 18**). We now explain that we use the *pearsonr* function from the SciPy Python package (ref. 1) to measure colocalization in a fully automated and completely unbiased manner. Please note that we deposited all primary microscopy data to the open-access data repository Zenodo, facilitating reanalysis of the data.

To address the concern regarding the expression levels of the glycosyltransferases, we have performed immunoblotting of fibroblasts from 2 patients and 2 healthy individuals (new **Supplementary Figure 8**). For most glycosyltransferases, we observed similar expression levels between patient and control fibroblasts. However, for some glycosyltransferases we observed large variation between the fibroblast lines, but these changes were not consistent between patients and controls. These changes therefore likely reflect true biological variation between individuals or fibroblast lines.

While we agree with the reviewer that the differences in Pearson correlation coefficients are generally small, PCC is independent of staining intensities and we observed consistent changes for all different glycosyltransferases that we investigated. Therefore, the

cumulative effect is likely much more important than the mislocalization of a single glycosyltransferase.

Moreover, mislocalization of glycosyltransferases is associated with other CDGs, including ATP6V0A2-CDG (2) and with defects in the COG complex (3,4). Simulations showed that already small effects in the localization of the glycosyltransferases can have a large impact on the final glycan product (5). In our revised manuscript, we now explain these points better and propose that the cumulative mislocalization of glycosyltransferases is a major contributor to the phenotype in the Stx5M55V patients (page 13).

- (1) Virtanen, Pauli, Ralf Gommers, Travis E. Oliphant, Matt Haberland, Tyler Reddy, David Cournapeau, Evgeni Burovski, et al. "SciPy 1.0: Fundamental Algorithms for Scientific Computing in Python." *Nature Methods* 17, no. 3 (March 1, 2020): 261–72. <https://doi.org/10.1038/s41592-019-0686-2>.
- (2) Kornak, Uwe, Ellen Reynders, Aikaterini Dimopoulou, Jeroen van Reeuwijk, Bjoern Fischer, Anna Rajab, Birgit Budde, et al. "Impaired Glycosylation and Cutis Laxa Caused by Mutations in the Vesicular H⁺-ATPase Subunit ATP6V0A2." *Nature Genetics* 40, no. 1 (January 2008): 32–34. <https://doi.org/10.1038/ng.2007.45>.
- (3) Pokrovskaya, Irina D., Rose Willett, Richard D. Smith, Willy Morelle, Tetyana Kudlyk, and Vladimir V. Lupashin. "Conserved Oligomeric Golgi Complex Specifically Regulates the Maintenance of Golgi Glycosylation Machinery." *Glycobiology* 21, no. 12 (December 1, 2011): 1554–69. <https://doi.org/10.1093/glycob/cwr028>.
- (4) Shestakova, Anna, Sergey Zolov, and Vladimir Lupashin. "COG Complex-Mediated Recycling of Golgi Glycosyltransferases Is Essential for Normal Protein Glycosylation." *Traffic* 7, no. 2 (2006): 191–204. <https://doi.org/10.1111/j.1600-0854.2005.00376.x>.
- (5) Jaiman, Anjali, and Mukund Thattai. "Golgi Compartments Enable Controlled Biomolecular Assembly Using Promiscuous Enzymes." Edited by Patricia Bassereau, Naama Barkai, and Alberto Luini. *ELife* 9 (June 29, 2020): e49573. <https://doi.org/10.7554/eLife.49573>.

Minor comments/suggestions

#1.12 - the last sentences of the introduction and the beginning of the discussion highlight the novelty of disease caused by a mutation in an alternative start site. While this may be the first described instance where loss of a short isoform due to change in an alternative start site causes disease, there are many previous descriptions of alternative start site variations leading to disease (see Bogaert et al, N-Terminal Proteoforms in Human Disease, 2020 for review)

We now discuss and cite the suggested review, and explain that while there are many previous descriptions of alternative start site variations leading to disease (ref. 1), our study reports the first known mutation in an alternative starting codon leading to human disease through the loss of a naturally occurring protein isoform.

- (1) Bogaert, Annelies, Esperanza Fernandez, and Kris Gevaert. "N-Terminal Proteoforms in Human Disease." *Trends in Biochemical Sciences* 45, no. 4 (April 1, 2020): 308–20. <https://doi.org/10.1016/j.tibs.2019.12.009>.

#1.13 - the images in the main figures are too small to evaluate the interpretation in the text for Figures 3-7. Insets are needed.

We thank the reviewer for this comment and now include inset panels for all relevant microscopy figures.

#1.14 - the description of SNARE complex formation in the first paragraph of the introduction is a bit hard to follow. Perhaps this can be clarified or schematized in a model.

We apologize if the explanation was inadvertently obfuscated and we have extensively rewritten this section (page 3).

#1.15 - the statement in the introduction that yeast Sed5p corresponds to Stx5 Long contradicts the authors' 2019 review. Please verify that the text is correct.

We hereby confirm that the text in the introduction is correct, and now explain in our revised manuscript that although Sed5p was originally believed to resemble mammalian Stx5S (ref. 1), it is now clear that it likely more resembles Stx5L, since an N-terminal COPI-binding tribasic motif has been identified in Sed5p (2). Please note that this latter study was published after our review.

- (1) Linders, Peter TA, Chiel van der Horst, Martin ter Beest, and Geert van den Bogaart. "Stx5-Mediated ER-Golgi Transport in Mammals and Yeast." *Cells* 8, no. 8 (August 2019): 780. <https://doi.org/10.3390/cells8080780>.
- (2) Gao, Guanbin, and David K. Banfield. "Multiple Features within the Syntaxin Sed5p Mediate Its Golgi Localization." *Traffic* 21, no. 3 (2020): 274–96. <https://doi.org/10.1111/tra.12720>.

#1.16 - more description and applicable references for CDG screening and how the results here compare to COG disorders would be helpful

We appreciate the question from the reviewer. CDG screening in general is performed by isoelectric focusing of transferrin or apolipoprotein CIII (the isoelectric point of transferrin is dependent only on the number of sialic acids) (ref. 1), but this only allows for interrogation of sialylation of these two serum proteins, respectively reflecting N-glycosylation and O-glycosylation. More modern methods of CDG diagnostics/investigations include mass spectrometry of intact transferrin (ref. 2) and so-called glycomics analyses of all N-glycans found in plasma (ref. 3). The advantage of these new methods is that they provide information on the exact glycan structure, not just on the number of sialic acids. COG disorders present with similar abnormalities of transferrin and apoCIII CDG screening and similar glycan structural abnormalities presenting with hypogalactosylation and hyposialylation. In addition, COG5-CDG presents with a similar increase in Man5 glycans as we measured here (Figure 1, refs. 4-7). For more information on the glycosylation patterns observed in COG disorders, we would like to refer the reviewer to our recent review (ref. 8) and we have added this explanation to the Results section on page 5.

- (1) Marklová, Eliška, and Ziad Albahri. "Screening and Diagnosis of Congenital Disorders of Glycosylation." *Clinica Chimica Acta* 385, no. 1 (October 1, 2007): 6–20. <https://doi.org/10.1016/j.cca.2007.07.002>.
- (2) Scherpenzeel, Monique van, Gerry Steenbergen, Eva Morava, Ron A. Wevers, and Dirk J. Lefeber. "High-Resolution Mass Spectrometry Glycoprofiling of Intact Transferrin for Diagnosis and Subtype Identification in the Congenital Disorders of Glycosylation." *Translational Research* 166, no. 6 (December 2015): 639-649.e1. <https://doi.org/10.1016/j.trsl.2015.07.005>.
- (3) Abu Bakar, Nurulamin, Dirk J. Lefeber, and Monique van Scherpenzeel. "Clinical Glycomics for the Diagnosis of Congenital Disorders of Glycosylation." *Journal of Inherited Metabolic Disease* 41, no. 3 (May 1, 2018): 499–513. <https://doi.org/10.1007/s10545-018-0144-9>.
- (4) Fung, C. W., G. Matthijs, L. Sturiale, D. Garozzo, K. Y. Wong, R. Wong, V. Wong, and J. Jaeken. "COG5-CDG with a Mild Neurohepatic Presentation." *JIMD Reports* 3 (2012): 67–70. https://doi.org/10.1007/8904_2011_61.
- (5) Paesold-Burda, Patricie, Charlotte Maag, Heinz Troxler, François Foulquier, Peter Kleinert, Siegrun Schnabel, Matthias Baumgartner, and Thierry Hennet. "Deficiency in COG5 Causes a Moderate Form of Congenital Disorders of Glycosylation." *Human Molecular Genetics* 18, no. 22 (November 15, 2009): 4350–56. <https://doi.org/10.1093/hmg/ddp389>.
- (6) Palmigiano, A., R. O. Bua, R. Barone, D. Rymen, L. Régál, N. Deconinck, C. Dionisi-Vici, et al. "MALDI-MS Profiling of Serum O-Glycosylation and N-Glycosylation in COG5-CDG." *Journal of Mass Spectrometry* 52, no. 6 (2017): 372–77. <https://doi.org/10.1002/jms.3936>.
- (7) Rymen, Daisy, Liesbeth Keldermans, Valérie Race, Luc Régál, Nicolas Deconinck, Carlo Dionisi-Vici, Cheuk-Wing Fung, et al. "COG5-CDG: Expanding the Clinical Spectrum." *Orphanet Journal of Rare Diseases* 7 (December 10, 2012): 94. <https://doi.org/10.1186/1750-1172-7-94>.
- (8) Linders, Peter T. A., Ella Peters, Martin ter Beest, Dirk J. Lefeber, and Geert van den Bogaart. "Sugary Logistics Gone Wrong: Membrane Trafficking and Congenital Disorders of Glycosylation." *International Journal of Molecular Sciences* 21, no. 13 (January 2020): 4654. <https://doi.org/10.3390/ijms21134654>.

#1.17 - To substantiate the claim that TGN46 levels are reduced (Figure 4h), this result should be quantified. If TGN46 levels are reduced, would you expect a more dramatic defect in Golgi architecture (Figure S5)?

As requested by the reviewer, we have quantified TGN46 levels based on the microscopy imaging (Supplementary Figure 11f). In addition, we performed comprehensive transmission electron microscopy experiments showing a dilation of both ER and Golgi cisternae (Figure 4a-c, gallery of images in Supplementary Figure 9) and we observed dilation of both the ER as Golgi cisternae.

-if there is a change in overall shape/staining of any component (GM130, TGN46, Stx5 overall) is comparing the Pearson Correlation Coefficient across cell types the appropriate way to analyze?

We agree that the distribution of not just the glycosyltransferases and Stx5, but of all Golgi markers is likely altered in Stx5M55V, especially considering the altered ultrastructure of the Golgi (see point #1.17). However, the main message of our analysis with the Pearson correlation coefficients is to show that the subcellular localizations of Stx5 and of all assessed glycosyltransferases relative to all assessed markers of *cis*- and *trans*-Golgi compartments is altered in the patient cells, thus an overall disorganization of the Golgi network. Please note that we believe that the Pearson correlation coefficient is the appropriate analysis to show this, because it is independent of staining intensities (and thus not affected by cellular protein levels). We now explain this at page 8 of our revised manuscript.

As discussed at point #1.11, we analyzed the microscopy images in a fully automated and unbiased fashion, and all primary microscopy data are deposited at the open-access repository Zenodo.

#1.19 - In 4C, why would there be a difference in overall b-COP staining if levels are unchanged (4g)?

As we now clarify in our revised manuscript (page 9), the reduced intensity of β COP in punctate structures possibly indicates reduced association with COPI vesicles.

#1.20 - 7d hard to believe this is really a significant difference

We now show the data distributions of all cells, which clearly show that this is a statistically significant difference.

#1.21 - For Figure 6, the legend title and title of 6c are confusing. I think this is supposed to highlight that Stx5L returns to the Golgi, so maybe 'Faster Golgi exit'? Also, if the rate of Golgi exit is faster for Stx5L, wouldn't we expect this value to be higher in 6d? It's not clear how this was quantified.

We have altered the title of Figure 6 into "Golgi exit" as suggested by the reviewer. Moreover, we now include the linear fits to Figure 6c (Figure 6b-c) to make it clear how we analyzed the data. The rates are negative because they indicate Golgi exit.

Reviewer #2 (Remarks to the Author):

This manuscript by Liners et al identifies a new mechanism of disease in humans, due to a pathogenic mutation in Stx5, an ER-Golgi SNARE protein. Affected individuals have a fatal multisystemic disease, liver failure and skeletal abnormalities, and abnormal glycosylation. The pathogenic variant in Stx5 is in an alternative starting methionine of syntaxin-5,

resulting in complete loss of the shorter protein isoform of syntaxin-5 (Stx5S), whilst the longer protein isoform (Stx5L) remains unchanged.

Linters et al undertake a detailed molecular investigation in patient fibroblasts and gene-edited HeLa cells, to determine the functional consequences of loss of Stx5S. A range of well-established molecular tools were utilised to study ER-Golgi trafficking (eg. RUSH assay, tsVSV-GFP trafficking) and determine transient protein interactions (FRET and FRET-FLIM). The authors demonstrated clear impairments in Golgi trafficking in these cell lines in some assays, and modest impairments in other endpoints. However, when all the results are considered collectively, it provides compelling evidence that the variant is likely pathogenic.

This manuscript is of interest to clinicians and geneticists, since it expands the clinical spectrum of genetic congenital disorders of glycosylation, and it suggests that mutations in Stx5 should be considered when screening patients with glycosylation abnormalities. Additionally, it is the first report for a mutation in an alternative starting methionine, causing protein isoform-specific effects.

I am happy that the authors have provided sufficient details in the methods sections to allow reproduction of their research. The statistical analysis is appropriate. The manuscript is well written, with appropriate background information, and a clear and concise discussion. The figures are clear, and well presented.

This manuscript would also have interest to the more general scientific field. This is because the authors also provide new evidence on the molecular function of Stx5S. Previously, it was thought that Stx5L was more critical to early Golgi trafficking, whilst Stx5S was more involved in late Golgi trafficking. Instead, the authors provide evidence that Stx5S is more important for anterograde trafficking, and loss of Stx5S in patient fibroblasts alters distribution of trafficking proteins and glycosylation enzymes. Additionally, loss of Stx5L indicated that Stx5L is more involved in retrograde trafficking. The authors also provide some additional information on the location-specific interaction of Stx5L and Stx5S with Bet1L, a known Golgi intra-trafficking protein. These additional insights would be of interest to researchers outside the human genetics field, as it provides fundamental insights into Stx5 protein function.

Paternal DNA was unavailable for genetic testing, and an unaffected sibling was also not tested. Given that Stx5 is a new gene and the affected individuals are homozygous for the variant, it would be good to know if the parents were consanguineous. If the parents are not consanguineous or not known to be, there would need to be some discussion about potential reasons for the variant being homozygous. Are the parents from the same small town or small religious group? Could it be the variant in the probands is hemizygous (ie one allele with the variant inherited from the mother and the gene partially or totally deleted in the paternally inherited allele)? It would be useful to include additional information about the ethnicity of the family.

We thank the reviewer for the positive assessment of our manuscript. We have investigated the possibility of consanguinity within the family (pedigree in Supplementary Figure 1), which we consider highly likely given the many homozygotic stretches including a long

stretch where the genetic variant resides (**Supplementary Table 4**). However, while we suspect that the paternal uncle (II:4) and maternal father (II:5) are the same person, we could not get confirmation from the family.

Reviewer #3 (Remarks to the Author):

NCOMMS-20-14459-T

*Congenital disorder of glycosylation caused by starting site specific variant in syntaxin-5
Peter Linders, Eveline Gerretsen, Angel Ashikov, Mari-Anne Vals, Natalia Revelo, Richard Arts, Melissa Baerenfaenger, Fokje Zijlstra, Karin Huijben, Kimiyo Raymond, Kai Muru, Olga Fjodorova, Sander Pajusalu, Katrin Ounap, Martin ter Beest, Dirk Lefeber, and Geert Bogaart*

In this study, Linders et al. discovered a mutation of the Golgi SNARE Syntaxin 5 as the cause of a novel and severe Congenital Disorder of Glycosylation (CDG), which is characterized by high abortion rate and badly skeletal dysplasia and abnormal facial features. Syntaxin 5 exists as two forms, long and short forms, by alternative translational initiation from the same mRNA. Interestingly, the mutation occurs at the second initiation codon, which blocks the production of the short form of Syntaxin 5. The authors showed that this unusual mutation induces mislocalization of several glycosyltransferases in the Golgi apparatus and thus leads to severe defects in protein glycosylation. They also perform cell biology assays to show trafficking defects in the mutant cells. Overall, the study is of high interest and physiologically relevant, but the quality of the data is poor, and the proposed molecular mechanism is not convincing. Specific points:

#3.1 - Figure 1d,e, can the different samples be analyzed in parallel and so one can directly compare the amount of each individual form?

These profiles reflect results from isoelectric focusing of serum transferrin and apolipoprotein CIII, which are both well-established techniques in the clinic diagnostic laboratory for CDG (refs. 1, 2). The relative ratios of the individual bands are interpreted for each individual sample and are used to interpret incompletely over fully glycosylated proteins for the diagnosis of CDG, as is common practice for CDG diagnosis. As such, the data of each sample is analyzed individually. We added the quantification of the individual lanes/samples in **Supplementary Table 2**. These data, together with the glycan mass spec data, therefore allow us to conclude that Stx5M55V is a CDG. We now explain this at **page 5** of our revised manuscript.

- (1) Scherpenzeel, Monique van, Gerry Steenbergen, Eva Morava, Ron A. Wevers, and Dirk J. Lefeber. "High-Resolution Mass Spectrometry Glycoprofiling of Intact Transferrin for Diagnosis and Subtype Identification in the Congenital Disorders of Glycosylation." *Translational Research* 166, no. 6 (December 2015): 639-649.e1. <https://doi.org/10.1016/j.trsl.2015.07.005>.
- (2) Marklová, Eliška, and Ziad Albahri. "Screening and Diagnosis of Congenital Disorders of Glycosylation." *Clinica Chimica Acta* 385, no. 1 (October 1, 2007): 6–20. <https://doi.org/10.1016/j.cca.2007.07.002>.

#3.2 - Figure 1g,h, since the y-axis is shown in a different scale, is there a difference in the amount of transferrin in control and patient serum?

The peak intensities of the mass spectrometry differ between the panels, because each graph shows an individual (patient or healthy control). The levels of circulating transferrin and serum proteins vary between individuals and the samples were obtained and processed at different times. As we now explain in our revised manuscript (page 5), the presence of many prominent peaks with incomplete glycosylation chains relative to peaks with complete glycosylation is a well-understood feature of CDG (ref. 1), allowing us to conclude that Stx5M55V is a novel CDG. The peak intensities in itself are not one-to-one related to the transferrin levels but also by the mass spectrometry conditions, and the isolation efficiency of transferrin from a certain sample.

- (1) Scherpenzeel, Monique van, Gerry Steenbergen, Eva Morava, Ron A. Wevers, and Dirk J. Lefeber. "High-Resolution Mass Spectrometry Glycoprofiling of Intact Transferrin for Diagnosis and Subtype Identification in the Congenital Disorders of Glycosylation." *Translational Research* 166, no. 6 (December 2015): 639-649.e1. <https://doi.org/10.1016/j.trsl.2015.07.005>.

#3.3 - Supplementary Table 3, what is the explanation of the differences between IV:9 and IV:9-10? Such a concern exists in multiple figures, which make the conclusions questionable.

IV:9 and IV:10 are different patients and although many clinical symptoms are similar, they are not identical, because of many different reasons (different genetic makeup, age, treatment, diet, etc.). Please note that, in contrast to experiments with genetically uniform cell lines and animals, high interindividual variation is commonly observed in studies with primary human material. We have included this comment on pages 8 and 13 of our revised manuscript.

#3.4 - Figure 2b,c, STX5 has been shown as an essential gene in yeast, are the patient fibroblasts viable?

We were indeed able to culture the fibroblasts normally and we did not detect differences in cell division and viability between control and patient cell lines. We now mention this at page 6.

#3.5 - Are these technical replicates or different cell lines?

We apologize to the reviewer for any confusion. Each lane in the Western blot in Figure 2b is a unique cell line; we use cell lines from 2 individual unique healthy donors and one cell line from each patient. We now clarify this in the legend of Figure 2, and for all the other figures where we show Western blot data.

#3.6 - The level of some proteins in the same cell type varies too much. What's the explanation of decreased Bet1L level in the patient fibroblasts?

We do not know the exact mechanism behind this observation but lower cellular levels of Bet1L have been demonstrated for certain COG defects (COG1, COG2, COG3) (refs. 1-3) and has been attributed to mislocalization to the ER where it is degraded (ref. 2). We have added this potential explanation to the Discussion section of our revised manuscript (page 13).

- (1) Shestakova, Anna, Elena Suvorova, Oleksandra Pavliv, Galimat Khaidakova, and Vladimir Lupashin. "Interaction of the Conserved Oligomeric Golgi Complex with T-SNARE Syntaxin5a/Sed5 Enhances Intra-Golgi SNARE Complex Stability." *The Journal of Cell Biology* 179, no. 6 (December 17, 2007): 1179–92. <https://doi.org/10.1083/jcb.200705145>.
- (2) Oka, Toshihiko, Daniel Ungar, Frederick M. Hughson, and Monty Krieger. "The COG and COPI Complexes Interact to Control the Abundance of GEARs, a Subset of Golgi Integral Membrane Proteins." *Molecular Biology of the Cell* 15, no. 5 (March 5, 2004): 2423–35. <https://doi.org/10.1091/mbc.e03-09-0699>.
- (3) Zolov, Sergey N., and Vladimir V. Lupashin. "Cog3p Depletion Blocks Vesicle-Mediated Golgi Retrograde Trafficking in HeLa Cells." *Journal of Cell Biology* 168, no. 5 (February 22, 2005): 747–59. <https://doi.org/10.1083/jcb.200412003>.

#3.7 - Figure 2d, the authors may want to increase DAPI intensity to make it more visible. Given that the lectin signal is normally not linear, a FACS analysis of the same cells would be more reliable.

We thank the reviewer for this suggestion and have increased the contrast of the DAPI channel in figures 2d and 2f. We have also performed the requested flow cytometry experiments with the lectin stainings. To achieve this, we performed lectin stainings on the patient fibroblasts and analysed them by flow cytometry (see figure below). Then, we compared histograms of the flow cytometry data with histograms of one representative microscopy experiment. As it was technically very challenging to consistently generate the large number of patient fibroblasts required for flow cytometry, and the flow cytometry data shows highly similar results to the microscopy data, we decided to continue analyzing the lectin data with microscopy. However, we now first perform a \log_{10} transformation on the intensity data before normalizing all data to the average of the control, to account for the non-linear lectin signal.

Side-by-side comparison of fluorescence intensity data of lectin stainings performed on Control (green) and StxM55V (orange) primary fibroblasts, by flow cytometry and microscopy.

#3.8 - Figure 3 and Supplementary Figure 4, the conclusion that “Glycosylation enzymes mislocalize in Stx5M55V patient fibroblasts” is invalid. Figure 3a, the morphology of the Golgi and the relative levels of the two colors are very different in the two cell lines, this will inevitably affect the quantitation results. Similar concerns exist in all following panels. The authors showed that different Golgi enzymes behave very differently, although they are all single-pass type II membrane proteins. What is the explanation? Does Stx5 bind some but not other enzymes? Figure 3c, the specificity of MAN2A1 is very poor, so the conclusion is not convincing. All proteins used in the study should be tested by Western blots as in Figure 2b.

A similar concern was raised by reviewer #1 (point #1.18). We have performed new EM experiments showing altered (more dilated) ultrastructure of the Golgi cisternae (new Figure 4, Supplementary Figure 9). Because of this altered Golgi structure, the distributions of not just glycosyltransferases and Stx5, but of all Golgi markers are likely altered in Stx5M55V cells. However, the main message of these experiments is to show that the subcellular localizations of Stx5 and of all assessed glycosyltransferases relative to all assessed markers of *cis*- and *trans*-Golgi compartments are altered in the patient cells, thus

an overall disorganization of the Golgi network. We now explain this at pages 8 and 9 of our revised manuscript.

We have also performed the requested Western blots of glycosyltransferases in new Supplementary Figure 8 (see also point #1.11). For most glycosyltransferases, we observed similar expression levels in patient and control fibroblasts. However, for some glycosyltransferases we observed large variation between the samples, reflecting differences between individuals or fibroblast lines. Nevertheless, these changes were not consistent between patient and control fibroblasts. Please note that we believe that the Pearson correlation coefficient is the appropriate analysis to analyze differences in colocalization between proteins, because it is independent of staining intensities and thus not affected by cellular protein levels. As discussed at point #1.11, we analyzed the data in a fully automated and unbiased fashion, and all microscopy data has been deposited to the repository Zenodo.

We have moved the data with MAN2A1 to the supplementary materials (Supplementary Figure 6), and confirmed our findings by performing additional experiments with another marker of the *trans*-Golgi (p230, new Supplementary Figure 7). As we now explain in our revised manuscript (Discussion section, page 13), we believe that the differential localization of glycosyltransferases and other Golgi proteins is a general defect caused by reduced retrograde Golgi-ER and intra-Golgi trafficking in Stx5M55V patient samples.

#3.9 - Supplementary Figure 5a, the conclusion that Stx5M55V does not affect the Golgi structure is not convincing. The morphology of the Golgi is very different between the two cell lines, different Golgi morphologies are also seen in Figure 3 and Supplementary Figure 4 even in different panels of the mutant cells. The EM image also indicated part of swollen Golgi cisternae is in a Stx5M55V patient cell. A gallery and quantification of the EM images should be included.

We thank the reviewer for this suggestion and we have now performed a more extensive EM investigation. We present representative micrographs and quantification in new Figure 4a-c and show a gallery of more EM images in new Supplementary Figure 9. We observed dilation of both Golgi and ER, which support the role for Stx5S in Golgi organization. Similar alterations in Golgi morphology have been observed in COG5 and COG7 deficient cells (ref. 1).

- (1) Oka, Toshihiko, Eliza Vasile, Marsha Penman, Carl D. Novina, Derek M. Dykxhoorn, Daniel Ungar, Frederick M. Hughson, and Monty Krieger. "Genetic Analysis of the Subunit Organization and Function of the Conserved Oligomeric Golgi (COG) Complex STUDIES OF COG5- AND COG7-DEFICIENT MAMMALIAN CELLS." *Journal of Biological Chemistry* 280, no. 38 (September 23, 2005): 32736–45. <https://doi.org/10.1074/jbc.M505558200>.

#3.10 - Figure 4, the quality of the images is of low quality due to the high background. The quality of the Western blots in Figure 4g,h is also very poor, with large variations within the same cell types (e.g. the two control lanes), raising a concern of the quality of the work.

We understand the comment of the reviewer, but please note that large variations between cells is inevitable when working with primary human material. The fibroblasts are primary cells derived from healthy donors and patients (with different genetic makeup, age, sex, diet, etc.). Compared to studies with genetically uniform cell lines and animal models, the variation is typically very large in human experiments but this variation is real and attributable to interindividual variation.

We have amended all figures by improving contrast and provide insets to make them clearer. Moreover, we now show loading controls with all Western blots (α -tubulin and GAPDH) that demonstrate similar amounts of protein were loaded for different samples.

To demonstrate the interindividual variation further, we isolated peripheral blood mononuclear cells (PBMCs) from four healthy individuals and directly immunoblotted them for Stx5 (new Supplementary fig. 4). After normalizing the band intensities of both Stx5L and Stx5S to the loading control GAPDH, it is clear that Stx5 expression varies vary substantially between individuals.

#3.11 - Figure 5, the functional assay is incomplete and not convincing. In addition to inhibiting ARF and COPI vesicle formation, BFA also modifies membrane lipids, and so the results does not provide much useful information. Although BFA-washout can be used to test anterograde trafficking. In this experiment, live cell imaging would be more convincing. For fixed cell imaging, multiple time points are needed. Additional Golgi and ER markers (e.g., PDI in Fig S6) should be included.

As suggested by the reviewer, we now include a BFA washout experiment using multiple timepoints. The results of this are presented in new Figure 5a-d and new Supplementary Figure 14a-c and show that loss of Stx5S compromises retrograde Golgi-ER trafficking, but not anterograde ER-Golgi trafficking. In addition, we now also include new experiments with temperature-synchronizable vesicular stomatitis virus G protein fused to GFP (VSVG-EGFP) in patient fibroblasts showing that whereas anterograde ER-Golgi trafficking is not affected in Stx5M55V cells, intra-Golgi and/or post-Golgi trafficking is reduced. The results of these VSVG-EGFP experiments are presented in new Figure 5e-g and Supplementary Figure 14d.

#3.12 - Figure 6 and 7, because of the inconsistent fluorescence intensity and the poor quality of images, the conclusion is invalid. This leaves no mechanism in this study.

We have revised Figures 6 and 7 and selected images with more comparable intensities. As we now discuss at page 17, all constructs were expressed at similar levels as estimated from the fluorescence intensities.

#3.13 - The author should perform rescue experiments in the cell based experiments by expressing Stx5S.

We applied several strategies to perform rescue experiments to supplement this study, but unfortunately, we were unsuccessful in doing so.

First, we attempted to rescue Stx5S expression in the patient fibroblasts by generating a lentivirally transduced stable cell line expressing Stx5S-V5 (and Stx5L-V5 or GFP-V5 expressing lines as controls). Unfortunately, we were unable to isolate single clones that had successfully integrated the V5-tagged constructs (see figure below, panel a).

Second, we attempted experiments where we co-expressed myc-tagged Stx5S together with VSVG-EGFP to rescue the transport of VSVG-EGFP in patient fibroblasts. This did not work however, as we did not manage to achieve workable expression levels of both Stx5S-myc and VSVG-EGFP in the same cell, likely due to CMV promoter silencing.

Third, we attempted transient overexpression by electroporation of GFP-tagged Stx5S fusion constructs in primary fibroblasts, followed by lectin stainings after 48 hours (see figure below, panels b-d). Unfortunately, we were not able to observe rescue of the glycosylation defect in the patient cells. Possibly, 48-hour incubation period is not long enough for all glycoproteins on the cell surface to have been recycled and all glycosylation routes to be fully rescued. A limitation of transient transfection is that plasmid expression strongly decreases after 48 hours and longer incubation times are not feasible. Thus, despite our efforts we have to conclude that we are unable to perform rescue experiments in the patient fibroblasts. This is likely a consequence of being unable to maintain the strict ratio of Stx5S over Stx5L expression required.

These results indicate that overexpression of Stx5S or Stx5L in HeLa cells also causes defects.

To remedy this, however, we generated a Stx5L-lacking fibroblast cell line from one of the control lines using CRISPR/Cas9 (Fib Stx5 Δ L). We then proceeded with the lectin stainings as described for the control and Stx5M55V patient fibroblasts and we observed a similar decrease in SNA-I labeling as compared to healthy control fibroblasts (new **Supplementary Figure 5b**). In contrast to Stx5M55V patient fibroblasts, we did not observe an increase in PNA labeling in the Stx5 Δ L fibroblasts. This difference must therefore be attributable to the loss of Stx5L over Stx5S, demonstrating that specifically Stx5S is necessary for physiological O-type mucin glycosylation. See also point #1.1 above.

- (a) Representative immunoblot for Stx5 on lysates of primary dermal fibroblasts from healthy donors (Ctrl), Stx5M55V patients (Stx5M55V) and Stx5M55V patient fibroblasts complement with GFP-V5, Stx5S-V5 or Stx5L-V5. GAPDH, loading control. No detectable rescue was observed.
- (b) Fibroblasts of healthy donors (green, ctrl) or Stx5M55V patients (Stx5M55V, orange) were transfected with either GFP or Stx5S-GFP (green in merge) and probed with PNA lectin (magenta in merge), Representative confocal micrographs. Scalebars, 10 μ m. DAPI in blue.
- (c) Quantification of (b). All data were \log_{10} -transformed and then normalized to the healthy donor. N = 151 (Ctrl + GFP), 124 (Ctrl + Stx5S-GFP), 111 (Stx5M55V + GFP)

and 86 (Stx5M55V + Stx5S-GFP) from 2 independent cell lines repeated twice. No changes were observed upon overexpression.

(d-e) Same as panels (b-c), but now for SNA-I lectin. N = 118 (Ctrl + GFP), 170 (Ctrl + Stx5S-GFP), 84 (Stx5M55V + GFP) and 85 (Stx5M55V + Stx5S-GFP) from 2 independent cell lines repeated twice.

Rescue attempts of VSVG-EGFP experiments as in **Figure 5 and Supplementary Figure 14**. Healthy donor fibroblasts (Ctrl, green) or Stx5M55V patient fibroblasts (Stx5M55V, orange) were co-transfected with pcDNAMycHis6 (+myc, empty vector control) or myc-tagged Stx5S (+Stx5S-myc) and VSVG-EGFP. Then the experiment was performed in the same way as for **Figure 5 and Supplementary Figure 14**. Myc was visualized with an anti-myc antibody (Cell Signaling Technology, #2272) followed by a donkey-anti-rabbit AF555 antibody. Representative epifluorescence micrographs are shown for timepoint $t = 60$ mins. Scalebars, 10 μ m. DAPI in dark blue. As demonstrated in this figure, it was technically not possible to obtain cells expressing both VSVG-EGFP and Stx5S-myc at sufficient levels for analysis. Therefore, we were unable to rescue the phenotype observed in the patient cells.

#3.14 - In many figures, different scales or scale bars are used in control vs. mutant cells. This is not acceptable.

We have amended all figures and they are now shown at the same magnification and with identical scale bars.

#3.15 - Some basic information is missing. For example, what antibodies are used for Syntaxin 5? Do the authors have an antibody specific for Stx5L?

We thank the reviewer for this question. We use antibodies recognizing both Stx5 isoforms and one antibody specific for Stx5L. All antibodies used are now listed in **Supplementary Table 5**.

#3.16 - There are typos and grammar mistakes in the text. A few example: page 9/37 line 221, "...was faster for Stx5L-SBP-mCitrine than for than for Stx5S or Stx5L Δ ER" should be "...was faster for Stx5L-SBP-mCitrine than that for Stx5S or Stx5L Δ ER". On page 9/37 line 230, "either without or with 30 mins biotin" should be either without or with biotin for 30 mins.

We apologize for the mistakes and we have corrected them in the text.

Reviewers' Comments:

Reviewer #1:

Remarks to the Author:

The authors have done an admirable job to address my main concern that defects observed are due to lower Stx5 levels overall, as opposed to isoform specific roles. It now appears that they are arguing that the ratio of isoforms is most important rather than isoform specific functions, which is consistent with data presented. The work appears ready for publication.

Reviewer #3:

Remarks to the Author:

The manuscript was improved, both technically and in writing. The result on the defective Golgi structure in Stx5M55V mutant cells is solid; however, the data supporting the trafficking defects and Golgi enzymes' mislocalization of Stx5 M55V is less convincing. The Pearson coefficient quantification was mostly different to a very small extent, even though the authors claimed the difference was statistically significant. Many results are still of low quality, and the newly enlarged images are of low resolution and thus are difficult to judge. The manuscript does not reach the standard of Nat Comm without addressing following concerns.

1. #3.8. In Fig. 3, sFig. 6 and sFig. 7, the authors stained fibroblasts of healthy donors or Stx5M55V patients by 12 pairs of Golgi proteins, 11 of which displayed decreased colocalization in cells with Stx5M55V mutation. Surprisingly, ZFPL1 and GALNT2 showed significantly better colocalization in Fig. 3, which contradicts the general idea that two Golgi proteins display less colocalizations with fragmented Golgi. This needs to be explained. Also in Fig. 3e, the Golgi morphology in Stx5M55V fibroblasts was quite different from that in Fig. 3a, c or g.

2. #3.11. In the new Fig. 5a, 6 min BFA is sufficient to inactivate ARF1/COPI but not long enough to allow the marker protein to go back to the ER, and a washout at this time point will not review any meaning results in trafficking. To test whether Stx5S deletion affects GALNT2 redistribution to the ER during BFA treatment, cells should be treated with BFA for varies times (0-60 min) followed by the detection of GALNT2 in the ER or Golgi. But if one wants to test ER-to-Golgi trafficking of GALNT2, one should treat cells with BFA for at least 1 hour to allow all GALNT2 move back to the ER, and then wash out BFA for different times. In Fig. 5b, how did the author get the shadowed line (error bars) between the four data points when cells were not imaged? It is not described anywhere that these results are from live cell imaging.

3. #3.13. It unfortunate that rescue experiment by three different methods all failed, and transient transfection with GFP-tagged Stx5S fusion did not correct the glycosylation defect. This is a key experiment as it confirms that the detected effects are specific to Stx5S mutation, given the complex genetic background of the patients. Alternatively, the author could repeat the IF experiment shown in Fig. 3 using the Stx5S-V5 and GFP-V5 cell lines that the authors already made by lentiviral transduction.

4. In Fig. 4d-g, Stx5L in fibroblasts from Stx5M55V patients displayed both stronger colocalizations with GM130 and PDI compared to that of healthy donors. Stx5L is known to be recycling between the ER and Golgi, so how could Stx5L localize better with both ER and Golgi at the same time? This should be explained.

Reviewer #3 (Remarks to the Author):

The manuscript was improved, both technically and in writing. The result on the defective Golgi structure in Stx5M55V mutant cells is solid; however, the data supporting the trafficking defects and Golgi enzymes' mislocalization of Stx5 M55V is less convincing. The Pearson coefficient quantification was mostly different to a very small extent, even though the authors claimed the difference was statistically significant. Many results are still of low quality, and the newly enlarged images are of low resolution and thus are difficult to judge. The manuscript does not reach the standard of Nat Comm without addressing following concerns.

#3.1 In Fig. 3, sFig. 6 and sFig. 7, the authors stained fibroblasts of healthy donors or Stx5M55V patients by 12 pairs of Golgi proteins, 11 of which displayed decreased colocalization in cells with Stx5M55V mutation. Surprisingly, ZFPL1 and GALNT2 showed significantly better colocalization in Fig. 3, which contradicts the general idea that two Golgi proteins display less colocalizations with fragmented Golgi. This needs to be explained. Also in Fig. 3e, the Golgi morphology in Stx5M55V fibroblasts was quite different from that in Fig. 3a, c or g.

We appreciate this remark from the reviewer, but please note that these data are in line with our conclusions: The electron microscopy images of Figure 4 (and Supplementary Figure 9) clearly show that the Golgi is not fragmented in Stx5M55V patients, but is instead enlarged (swollen) and consists of continuous Golgi cisternae. Considering the role of intra-Golgi transport for the recycling and proper localization of Golgi resident enzymes, we attribute the mislocalization of ZFPL1 and GALNT2 to the improper transportation of these enzymes to the correct compartment. In line 221 we now clearly mention that the Golgi is not fragmented in the Stx5M55V patients and we apologize for the confusion.

We have amended Figure 3e to now include a more representative image of Stx5M55V fibroblasts.

#3.2 In the new Fig. 5a, 6 min BFA is sufficient to inactivate ARF1/COPI but not long enough to allow the marker protein to go back to the ER, and a washout at this time point will not review any meaning results in trafficking. To test whether Stx5S deletion affects GALNT2 redistribution to the ER during BFA treatment, cells should be treated with BFA for varies times (0-60 min) followed by the detection of GALNT2 in the ER or Golgi. But if one wants to test ER-to-Golgi trafficking of GALNT2, one should treat cells with BFA for at least 1 hour to allow all GALNT2 move back to the ER, and then wash out BFA for different times. In Fig. 5b, how did the author get the shadowed line (error bars) between the four data points when cells were not imaged? It is not described anywhere that these results are from live cell imaging.

As suggested by the reviewer, we performed experiments with longer incubation times of BFA. As we now show for 15 min incubation in new Supplementary Figure 14e-f, we still observed incomplete ER localization of GALNT2 for Stx5M55V patient fibroblasts. Since the differences between fibroblasts from Stx5M55V patient and healthy subjects is similar for the long and short (6 min) incubation times, these new data strengthen our conclusion that Stx5S mainly

plays a role in retrograde Golgi-ER trafficking. These new data are mentioned at line 257 in our revised manuscript.

#3.3 It unfortunate that rescue experiment by three different methods all failed, and transient transfection with GFP-tagged Stx5S fusion did not correct the glycosylation defect. This is a key experiment as it confirms that the detected effects are specific to Stx5S mutation, given the complex genetic background of the patients. Alternatively, the author could repeat the IF experiment shown in Fig. 3 using the Stx5S-V5 and GFP-V5 cell lines that the authors already made by lentiviral transduction.

Unfortunately, we were still unable to obtain successfully transduced cells (i.e., expressing Stx5S-V5) despite new attempts. As previously, we encountered difficulties in culturing cells after exposure to lentiviral particles as these cells stopped dividing. As observed in the Western blot of cell lysates from these cells (see Figure 1 below), we were not able to achieve expression of Stx5S-V5. This is likely a consequence of a cellular requirement to have roughly equimolar expression of Stx5S and Stx5L, as suggested by the equimolar expression of the Stx5 isoforms in PBMCs from healthy donors in Supplementary Figure 4. This is also supported by our findings that both the absence of Stx5S and Stx5L result in glycosylation defects (see Figure 2 and Supplementary Figure 5). Thus, we were unfortunately still unable to conduct rescue experiments, likely because the ratio of the two isoforms is critical.

Figure 1: No observed expression of Stx5S-V5 in lentivirally transduced fibroblasts.

#3.4 In Fig. 4d-g, Stx5L in fibroblasts from Stx5M55V patients displayed both stronger colocalizations with GM130 and PDI compared to that of healthy donors. Stx5L is known to be recycling between the ER and Golgi, so how could Stx5L localize better with both ER and Golgi at the same time? This should be explained.

A possible explanation for the increased localization of Stx5L in both the Golgi and the ER in Stx5M55V fibroblasts is that Stx5L can compensate for the loss of Stx5S and is therefore more present at the Golgi. Due to the recycling nature of Stx5L, this in turn might result in more Stx5L being present at the ER, while it is displaced from trafficking intermediates and post-Golgi compartments. In support of this, the Stx5L staining is highly dispersed in healthy donor fibroblasts, while Stx5L is present in a more defined Golgi-like structure in Stx5M55V fibroblasts

(see Figure 4d-g). We have included this explanation at lines 248-250 in the revised version of our manuscript.